# Between belief and fear - Reinterpreting prone burials during the Middle Ages and early modern period in German-speaking Europe

Amelie Alterauge[1,2], Thomas Meier[2], Bettina Jungklaus[3], Marco Milella[1], Sandra Lösch[1]*

1 Department of Physical Anthropology, Institute of Forensic Medicine, University of Bern, Bern, Switzerland, 2 Institute for Pre- and Protohistory and Near Eastern Archaeology, University of Heidelberg, Heidelberg, Germany, 3 Anthropologie-Büro Jungklaus, Berlin, Germany

* sandra.loesch@irm.unibe.ch

**Data Availability Statement:** All relevant data are within the paper and its Supporting Information files.

## Abstract

Prone burials are among the most distinctive deviant burials during the Middle Ages and early modern period. Despite their worldwide distribution, the meaning of this burial practice is still a matter of debate. So far, a comprehensive analysis of prone burials is lacking for Central Europe. By compiling evidence from Germany, Switzerland and Austria, this study investigates how these findings fit into the scope of medieval funerary practices. 95 prone burials from 60 archaeological sites were analyzed regarding geographical distribution, dating, burial features, body position, age-at-death and sex. We applied descriptive statistics accompanied by multiple correspondence analysis in order to highlight possible multivariate patterns in the dataset. Prone burials occur in funerary and non-funerary contexts, with a predominance of single churchyard burials, followed by favored and exterior location and settlements. In terms of grave features, the majority of churchyard burials do not differ from regular graves. Multivariate patterns appear to reflect diachronic changes in normative burial practices. We found a significant correlation between burial location and dating, due to a higher frequency of high medieval males in favored locations. In these cases, prone position is interpreted as a sign of humility, while similar evidences from late and post-medieval times are seen as an expression of deviancy. Apparent lack of care during burial reveals disrespect and possible social exclusion, with inhumations outside consecrated ground being the ultimate punishment. In some regions, apotropaic practices suggest that corpses should be prevented from returning, as attested in contemporaneous sources and folk beliefs. We hypothesize that the increase of prone burials towards the late and post-medieval period is linked to such practices triggered by epidemic diseases. The multiplicity of meanings that prone position might have in different contexts demands for careful interpretations within the same regional and chronological frame.

**Funding:** The author(s) received no specific funding for this work.

**Competing interests:** The authors have declared that no competing interests exist.

## Introduction

Atypical burials are characterized by a range of features, such as burial location, position and/or grave goods, deviating from what is usually observed for a specific geographical and chronological context [1]. Deviancy, if interpreted as deviation from a norm, depends upon a society's social norms and may vary between different times, regions and even sites. The use of this term is problematic since it is based on the dichotomy between 'normal' (or regular/typical) and 'abnormal' (or irregular/atypical), even though archaeological cultures usually have a broad range of funerary practices [2, 3]. In particular, the term has a negative connotation that archaeologists instinctively transfer from the burial to the individual during life. Such deviancy might have originated from otherness like disability, profession, provenance, or religion perceived as 'odd' by fellows; those social outcasts potentially required special funerary provisions [4]. Additionally, circumstances of sudden death not in accordance with society's expectations and not allowing for the normative rituals of dying (e.g. suicide, execution, drowning, death outdoors, etc.) were possible reasons for deviancy [5]. As demonstrated by Shay [6], differential treatment in burial is not necessarily only due to negative perceptions surrounding the deceased but may also express some kind of 'positive deviancy'.

In this contribution, we will use the term 'deviant' burial, following the majority of topic-specific publications [4, 7, 8]. We will, however, perceive it as equivalent to the German 'Sonderbestattung', describing diverging burial practices without any qualitative connotation [1, 9].

Deviant burials have been intensively studied in archaeology because of their relative rarity and enigmatic appearance [7, 10]. The study of these findings has a long tradition in British archaeology [4], with an increasing focus on their possible social meaning. In contrast, in Continental Europe the cultural and social relevance of deviant burials are rarely addressed, and their analysis is limited to descriptive case studies and comparisons with other isolated findings [11, 12]. In consequence, their interpretation has often relied on similarly isolated analogies.

In Europe, prone burials are among the most distinctive types of deviant burials during the Middle Ages and in most cases required a deliberate decision of the burial party to place a body face-down. Medieval burial customs, at the latest since the Christian faith prevailed in Central Europe, are typically represented by single graves on a shared burial ground in extended supine position [13, 14]. They are usually oriented West-East, with the face looking East in order to see Christ resurrecting in Jerusalem, as it was demanded by the Catholic and Orthodox and later also the Protestant Church. Burial norms were fixed by ecclesiastical laws defining any aberrations as deviant, which is still influential on today's perception of non-normative burials. Grave goods, which had largely disappeared from the early 7th century onwards in Southern and Western Europe, became more frequent again during the High Middle Ages (11th-13th centuries AD) in form of individual items like coins, rings, pilgrims' badges, crosses, papal seals, etc. with probable symbolic value [15–17]. From the 15th century onwards, elements of clothing re-entered the archaeological record. In post-medieval times, religious as well as profane burial goods and elements of clothing were witnesses of funerary rites (e.g. laying-out), folk beliefs or religious denomination [18–20]. In the advanced Early Middle Ages (8th-10th centuries AD), the idea of an "ideal way of dying" developed first in monastic communities, but was quickly adopted by the nobility and–at the latest at the turn of the first millennium–by the lower classes as well [21–24]. The individual should have enough time to prepare him- and herself for death, to repent, to distribute his/her possessions and to receive the last rites; death should ideally occur among family and friends.

The value of medieval burials as a source of information about past social variability was questioned up until the 2000s. The reservation about their use for social reconstruction stems from assumed homogeneity, and from the monopoly of church on funerary rituals [25]. Most studies have therefore only briefly considered medieval deviant burial practices, and without placing them in a wider interpretive frame [26]. The lack of a comprehensive analysis of deviant burials in Continental Europe for the Middle Ages and (early) modern period (11th-19th centuries AD) has so far hampered any discussion of these burials on a larger scale.

## Deviant and prone burials in European pre- and protohistory: An overview

For Europe, Murphy [7] and Reynolds [4] have so far provided the most extensive overviews on deviant burials and how their definition is influenced by cultural notions of the respective context but also by elements of folklore and superstition. Numerous studies have dealt specifically with prehistoric [27], Roman [28, 29], medieval [30–35], Anglo-Saxon [4] and Viking age [36, 37] deviant burials in various geographical contexts. Most research in Britain, particularly England, was focused on the Roman or Anglo-Saxon period [4, 7, 28, 50], whilst late medieval funerary variability received less attention and has only recently been tackled [38, 39]. In Christian burial grounds, prone burials obviously stand out as different regarding the manner of burial but were apparently still deemed suitable for the inclusion in a Christian community's cemetery [39].

The aforementioned studies have in common that they did not exclusively deal with prone burials but also with other forms of deviancy, e.g. diverging orientation, side or crouched position, decapitation, and mutilation or fixation (e.g. stoning, nailing). The published examples were interpreted in the context of social stigmatization, exclusion, and/or postmortem punishment of the deceased.

Arcini [40, 41] compiled the first review of published prone burials including over 600 individuals from different world regions from prehistory to modern times. She interpreted the large geographical and chronological distribution of prone burials as a cross-cultural phenomenon with potentially shared intention. As Arcini [40, 41] had exemplarily shown, individuals who deviated from society's norms in different cultures were reserved the indignity of being buried face down.

Similarly, according to Wilke [42] and Kyll [43], prone burials were not careless disposals, but intentional acts of burying the dead. Wilke [42] alleged that this practice was intended to prevent the return of 'dangerous' dead to the world of the living. Burying a corpse with the face down would have not allowed the soul to escape the ground or to get back into the mouth [43]. In addition, prone position was believed to ward off epidemic diseases which would otherwise spread from the deceased to the living [44].

Deviant (including prone) medieval burials in Eastern and Southeastern Europe have been commonly attributed to *vampires*, based on a comparison between archaeological data and historical and ethnological sources. The cultural figure of the vampire, a version of the reanimated corpse, can be traced in written sources of the Balkans and Eastern European regions as early as the 11th century. The belief, originally connected to pagan spiritualism, spread after the introduction of Christianity inhumation as main burial practice [45]. In medieval Western Europe, however, revenants mainly appeared to their fellows in visions and dreams and were usually acting more friendly and physically less threatening [46]. Interestingly, no medieval source documents prone burial as a mean to ban revenants [47]. Alternative explanations for deviant burials, such as judicial practices, have only recently been suggested [48, 49].

Prone burials in early medieval Southern Germany (5th-7th centuries AD) are also interpreted as a protective practice against dangerous dead [35, 50]. Following the work of Philpott [51] on Roman Britain, Walter [50] defined three categories of prone burials, attributing a specific meaning to each of them. These included: prone individuals with no physical peculiarities (e.g. trauma, mutilation), burials with physical peculiarities, and prone inhumations in double burials with one prone and one supine individual. According to Walter [50], the first group would include individuals discriminated on the basis of their specific circumstances of death; the second corresponded to disabled individuals or executed criminals. The third would represent sexually "deviant" individuals such as homosexuals.

Isolated examples of deviant and prone burials from medieval and early modern Germany have also been briefly discussed [32, 52, 53]. In summary, also for Central Europe, prone burials are mainly interpreted as a mean to disempower dangerous dead, similar to what is proposed for other regions. However, this interpretation, although particularly fascinating for the public [54, 55], is still lacking a critical theoretical basis.

In this study, we aim to investigate how and if prone burials fit into the scope of medieval and post-medieval funerary practices. Based on prone burials from funerary and non-funerary contexts, we analyze their occurrence, frequency and appearance in Western Central Europe. We expect that their geographical and chronological distribution reveals patterns related to the interpretation of these burials. By doing so, we close a research gap that exists for medieval Europe regarding prone burials in particular and deviant burials in general.

## Material and methods

### Study area

The focus of this study are prone burials from the German-speaking countries Germany (D), Switzerland (CH) and Austria (AT). This geographical area was selected due to overall shared language and similar cultural history (e.g. former Holy Roman Empire, Reformation, Thirty Years War), and for being largely underrepresented in modern research on post-Roman deviant burials. Prone burials from the francophone Swiss cantons Vaud, Valais and Fribourg were also included in our sample.

Our research was limited to prone burials post-dating 950 AD, since we assume that specific burial norms, such as the use of churchyards as burial grounds, were fully established in the Frankish and early German Empire by that date [56]. For the same reason, cases from Northeastern Europe were considered only from the 12th century onward.

### Data collection

Cases of prone burials were retrieved after a comprehensive review of local archaeological publications and excavation gazetteers. Additionally, documentation for unpublished cases was obtained from archaeologists and local cultural heritage institutions. This study exclusively deals with archaeological skeletal material, and all necessary permits were obtained for the study, which complied with all relevant regulations. The remains are stored in the respective archives of the heritage institutions in charge. For Germany and Austria, those are the State Heritage or Monument Protection Departments, for Switzerland, those are the Cantonal Archaeological Services. Details on storage location can be found in the cited publications.

Prone inhumations were included in this study if they: a) dated following 950 AD; b) were part of burials including a maximum of three individuals; c) information on age-at-death and sex of the individual were available. Multiple (including more than three individuals) burials are often the result of catastrophic events leading to random or necessarily careless deposition of the dead [57–59]. They would potentially bias the variability of our sample and were not

considered. The same applies to execution sites where prone burials are regularly observed [60–62]. However, the relative abundance of prone positions in the aforementioned burial contexts must be kept in mind when it comes to interpretation.

After the above screening, our sample includes 95 prone burials from 60 archaeological sites (Table 1, S1 File). We classified the burials according to eleven categorical variables (Table 2), chosen in order to summarize their funerary and demographic features and to maximize their comparability while minimizing the bias introduced by lax, unclear or missing information. Information on specific arm positions, categories of grave goods, specific age-at-death classes, and pathological conditions were not included due to an overall lack of pertinent data or due to dubious attributions. We differentiated between adults (≥20 years) and sub-adults. For the descriptive results, we worked with the published age estimations and categorized them into age classes (I = infant (0–12 years), J = juvenile (13–19 years), YA = young adult (20–39 years), MA = middle adult (40–59 years), OA = old adult (above 60 years)).

In order to increase the size of our sample, seven previously undated prone inhumations were radiocarbon dated at the LARA laboratory at the Department of Chemistry and Bio-chemistry at the University of Bern [63, 64]. Sampling was permitted by the heritage state agencies in charge. In addition, the radiocarbon dates of two so far unpublished specimens are also presented (Table 3).

### Data analysis

Geographical and chronological frequencies of prone burials and of each of the eleven chosen variables were first calculated in order to explore the overall variability of our sample. Possible associations between variables were further analyzed by means of a Fisher's Exact Test with exact calculations of *m x n* matrices [65].

In a second step, we analyzed our dataset by means of a multiple correspondence analysis (MCA). MCA is an ordination method suitable for exploring the possible presence of multi-variate patterns in a categorical dataset [66]. A multivariate set is reduced to a limited number of dimensions, which can be used to visualize the relative similarity between cases as their Euclidean distance in (typically) bivariate plots. When performing MCA, missing data were handled by using the sample mode for each variable.

All analyses were performed with IBM SPSS® Statistics 26.0. Results of MCA were further visualized in JMP 15.10 (SAS Institute 2019). For all tests alpha was set at 0.05.

## Results

### Geographical and chronological distribution

Our sample includes 76 burials from Germany, 16 from Switzerland and three from Austria (Fig 1, Table 1). Evidently, the regional distribution is highly biased and reflects above all the research areas of the authors. Compared to the size of the countries under study, prone burials are overrepresented in Swiss cemeteries and underrepresented in Austria. Prone burials are more common in western Switzerland than in the east, leaving a blank spot until the area of the Inn River.

In Germany, the distribution is heterogeneous, with the states of Brandenburg and Mecklenburg-West Pomerania showing the highest frequencies of prone burials and the states of Lower Saxony, Saxony-Anhalt and Saxony the lowest. A connection between the West and East might be suggested along the German Mittelgebirge and along the Main River, possibly functioning as some sort of communication corridor.

The chronological distribution of the sample is as follows: 19 prone burials (20%) date to the High Middle Ages (10th-13th centuries), 31 (32.7%) to the Late Middle Ages (13th-16th

**Table 1. Prone burials integrated in this study.**

| No. | Site | Grave | State/Country | Period | Burial type | Burial place | Burial location | Burial container | Arm position | Arm location | Leg position | Orientation | Grave goods | Sex | Age1 | Age2 | Reference (S1 File) |
|---|---|---|---|---|---|---|---|---|---|---|---|---|---|---|---|---|---|
| 1 | Altlichtenwarth, Kirche | 35/2 | 3/AT | 1 | single | funerary | churchyard | | regular | front | extended | West | 0 | male | adult | YA | Grossschmidt 2014; Sauer 2014 |
| 2 | Anklam, Marienkirchhof | 32 | MV/DE | 4 | single | funerary | churchyard | | regular | front | extended | West | | indet. | adult | A | Weber 1999 |
| 3 | Anklam, Pferdemarkt | 92 | MV/DE | 5 | single | funerary | exterior | | disordered | | flexed | West | | indet. | adult | A | Museum im Steintor 2009 |
| 4 | Bayreuth, Stadtkirche | 55 | BY/DE | 3 | single | funerary | churchyard | | | | extended | West | 0 | indet. | subadult | I | Wintergerst 2013 |
| 5 | Bayreuth, Stadtkirche | 118 | BY/DE | 3 | single | funerary | churchyard | | regular | | extended | West | 0 | male | adult | A | Wintergerst 2013 |
| 6 | Bayreuth, Stadtkirche | 235 | BY/DE | 3 | single | funerary | churchyard | | regular | | extended | West | 0 | indet. | adult | A | Wintergerst 2013 |
| 7 | Belfaux, Pré Saint Maurice | 475 | FR/CH | 4 | single | funerary | exterior | | regular | front | extended | Northeast | 1 | female | adult | YA | McCullough pers. comm. |
| 8 | Berlin, Petriplatz | 4627 | BE/DE | 3 | single | funerary | churchyard | shroud | | | | West | | indet. | subadult | J | Melisch 2017 pers. comm. |
| 9 | Berlin, Petriplatz | 4806 | BE/DE | 1 | single | funerary | churchyard | shroud | regular | front | extended | West | 0 | male | adult | OA | Melisch 2017 pers. comm. |
| 10 | Berlin, Tempelhofkirche | 2 | BE/DE | 1 | single | funerary | favored | | disordered | front | extended | West | 0 | male | adult | MA | Heinrich 1954 |
| 11 | Borkum, Walfängerfriedhof | 146 | NI/DE | 3 | double | funerary | churchyard | | | | | West | 0 | male | adult | YA | Burkhardt 2017a, 2017b |
| 12 | Borkum, Walfängerfriedhof | 147 | NI/DE | 3 | double | funerary | churchyard | | | | | West | | male | adult | OA | Burkhardt 2017a, 2017b |
| 13 | Bülach, Rathausgasse 1 | 52 | ZH/CH | 4 | single | funerary | churchyard | | | | extended | Northwest | 0 | male | adult | YA | Bader/Langenegger 2013 |
| 14 | Bülach, Rathausgasse 1 | 86 | ZH/CH | 4 | single | funerary | churchyard | coffin | disordered | front | | Northwest | 0 | female | adult | YA | Bader/Langenegger 2013 |
| 15 | Büren a. d. Aare, Chilchmatt | 91 | BE/CH | 1 | single | funerary | churchyard | shroud | regular | front | | Northwest | 0 | female | adult | MA | Eggenberger et al. 2019 |
| 16 | Diepensee, Friedhof | 278 | BB/DE | 1 | single | funerary | churchyard | | | | | West | 0 | male | subadult | J | Jungklaus pers. comm.; Jungklaus 2008, 2009 |
| 17 | Diepensee, Friedhof | 379 | BB/DE | 1 | single | funerary | churchyard | | | | | West | | male | adult | MA | Jungklaus pers. comm.; Jungklaus 2008, 2009 |
| 18 | Echenbrunn, Mühlenweg | 725 | BY/DE | 3 | single | funerary | churchyard | | regular | front | | West | 0 | male | adult | A | Seidel/Bohnet 2018 |
| 19 | Echenbrunn, Mühlenweg | 788 | BY/DE | 3 | double | funerary | churchyard | coffin | regular | front | extended | West | 0 | female | adult | A | Seidel/Bohnet 2018 |
| 20 | Echenbrunn, Mühlenweg | 789 | BY/DE | 3 | double | funerary | churchyard | coffin | | | | | | indet. | subadult | I | Seidel/Bohnet 2018 |
| 21 | Elten, Stiftskirche | 33 | NW/DE | 1 | single | funerary | favored | coffin | regular | front | extended | West | 0 | female | adult | YA | Binding 1970; Jungklaus 1970 |
| 22 | Elten, Stiftskirche | 34a | NW/DE | 1 | double | funerary | favored | coffin | regular | front | extended | West | 0 | indet. | adult | YA | Binding 1970; Jungklaass 1970 |
| 23 | Elten, Stiftskirche | 34b | NW/DE | 1 | double | funerary | favored | coffin | regular | front | extended | East | 0 | male | adult | MA | Binding 1970; Jungklaass 1970 |
| 24 | Erding, Melkstatt | 3b/1977 | BY/DE | 5 | single | funerary | exterior | | | | | Southwest | 1 | female | adult | MA | Maier 1980, 1981, 1988 |
| 25 | Erding, Melkstatt | 1/1981 | BY/DE | 5 | single | funerary | exterior | | regular | back | extended | West | | indet. | adult | A | Maier 1980, 1981, 1988 |

(*Continued*)

**Table 1.** (Continued)

| No. | Site | Grave | State/Country | Period | Burial type | Burial place | Burial location | Burial container | Arm position | Arm location | Leg position | Orientation | Grave goods | Sex | Age1 | Age2 | Reference (S1 File) |
|---|---|---|---|---|---|---|---|---|---|---|---|---|---|---|---|---|---|
| 26 | Erding, Melkstatt | 4/1981 | BY/DE | 5 | single | funerary | exterior | | regular | | extended | East | | indet. | adult | A | Maier 1980, 1981, 1988 |
| 27 | Erding, Melkstatt | 6/1981 | BY/DE | 5 | single | funerary | exterior | | disordered | | extended | East | | indet. | adult | A | Maier 1980, 1981, 1988 |
| 28 | Esslingen, St. Dionysius | IIIc-h ab 103 | BW/DE | 3 | single | funerary | churchyard | | | | | West | 0 | female | adult | YA | Fehring/Scholkmann 1995; Francken 2019 pers. comm. |
| 29 | Flintsbach/Inn, St. Peter am Madron | 230/519 | BY/DE | 1 | single | funerary | favored | | | | | West | 1 | male | adult | MA | Meier 2002; Meier 2015 |
| 30 | Flintsbach/Inn, St. Peter am Madron | 84 | BY/DE | 2 | triple | funerary | churchyard | | regular | | | West | 0 | female | subadult | I | Mohr et al. 2001 |
| 31 | Flintsbach/Inn, St. Peter am Madron | 630 | BY/DE | 1 | single | funerary | churchyard | | | | extended | West | 0 | indet. | adult | A | Meier 2020 pers. comm. |
| 32 | Flintsbach/Inn, St. Peter am Madron | 660 | BY/DE | 1 | single | funerary | churchyard | | | | | West | 0 | male | adult | YA | Meier 2020 pers. comm.; Lösch 2009 |
| 33 | Flintsbach/Inn, St. Peter am Madron | 666 | BY/DE | 1 | single | funerary | churchyard | | regular | front | | West | 0 | male | adult | OA | Meier 2020 pers. comm.; Lösch 2009 |
| 34 | Flintsbach/Inn, St. Peter am Madron | 805 | BY/DE | 1 | single | funerary | churchyard | | | | | West | 0 | indet. | adult | A | Meier 2020 pers. comm. |
| 35 | Freiburg, Münsterplatz | 30 | BW/DE | 3 | double | funerary | churchyard | coffin | regular | front | extended | East | 1 | male | adult | A | Bohnet 2018 pers. comm. |
| 36 | Freiburg, Münsterplatz | 198 | BW/DE | 3 | single | funerary | churchyard | coffin | | | | West | 1 | indet. | adult | A | Jenisch/Bohnet 2015 |
| 37 | Füssen, Magnusplatz | 34 | BY/DE | 3 | single | funerary | churchyard | | regular | front | | Southwest | 0 | indet. | adult | YA | Wintergerst 2015 |
| 38 | Grabow, Kirchenplatz | 4 | MV/DE | 5 | single | funerary | churchyard | | disordered | side | extended | West | | male | adult | A | Schulze 2015 |
| 39 | Greifswald, Kloster Eldena | 73 | MV/DE | 5 | single | funerary | churchyard | | regular | front | | South | 0 | male | adult | A | Kaute 2011a, b |
| 40 | Greifswald, St. Jacobikirchhof | 159 | MV/DE | 3 | double | funerary | churchyard | coffin | | | | West | | male | adult | A | Ansorge 2003 |
| 41 | Hanau-Kesselstadt, Friedenskirche | 6 | HE/DE | 5 | single | funerary | churchyard | | | | extended | South | 0 | male | adult | YA | Jüngling 2004 |
| 42 | Hanau-Kesselstadt, Friedenskirche | 75 | HE/DE | 5 | single | funerary | churchyard | coffin | regular | front | extended | East | 0 | female | adult | YA | Jüngling 2004 |
| 43 | Hanau-Kesselstadt, Friedenskirche | 85 | HE/DE | 5 | single | funerary | churchyard | coffin | disordered | side | extended | West | 1 | male | adult | OA | Jüngling 2004 |
| 44 | Hanau-Kesselstadt, Friedenskirche | 88 | HE/DE | 4 | single | funerary | churchyard | | regular | front | extended | East | 0 | indet. | subadult | J | Jüngling 2004 |
| 45 | Harsefeld, Kloster | 5 | NI/DE | 3 | single | funerary | favored | coffin | | | | West | | male | adult | A | Nösler 2014 |
| 46 | Klein Hoym, Friedhof | 31518 | ST/DE | 2 | single | funerary | churchyard | | regular | front | flexed | Northwest | 0 | female | adult | A | Selent 2018 |
| 47 | Klein Hoym, Friedhof | 31610 | ST/DE | 2 | single | funerary | churchyard | | regular | front | extended | Northwest | 0 | female | adult | A | Selent 2018 |
| 48 | Konstanz, Heiliggeist Spital | 867 | BW/DE | 2 | single | funerary | churchyard | | | | other | West | 0 | male | adult | A | Berszin 1999 |
| 49 | Konstanz, Petershausen | 505 | BW/DE | 5 | single | funerary | churchyard | | disordered | front | extended | West | 0 | female | adult | OA | Berszin 2009 |
| 50 | Konstanz, Petershausen | 588 | BW/DE | 1 | single | funerary | churchyard | | | | | West | 0 | female | adult | YA | Berszin 2009 |
| 51 | Lausanne Vidy, CIO | 1558 | VD/CH | 3 | single | funerary | churchyard | coffin | regular | front | | West | | male | adult | OA | Guichon et al. 2017 |

(*Continued*)

**Table 1.** (Continued)

| No. | Site | Grave | State/Country | Period | Burial type | Burial place | Burial location | Burial container | Arm position | Arm location | Leg position | Orientation | Grave goods | Sex | Age1 | Age2 | Reference (S1 File) |
|---|---|---|---|---|---|---|---|---|---|---|---|---|---|---|---|---|---|
| 52 | Luppa | 1 | SH/DE | 4 | single | non-funerary | settlement | | regular | front | extended | East | 0 | female | adult | YA | Häckel 2009, 2012 |
| 53 | Münster, Domherrenfriedhof | 405 | NW/DE | 5 | single | funerary | churchyard | coffin | disordered | back | | East | 0 | male | adult | OA | Schneider et al. 2011 |
| 54 | Münster, Jüdefelderstrasse | 5507–2 | NW/DE | 5 | single | non-funerary | exterior | | regular | front | extended | North | 0 | male | adult | YA | Thier 2017 |
| 55 | Münster, Stubengasse | 417 | NW/DE | 5 | single | funerary | exterior | | | | | East | 0 | male | adult | A | Winkler 2008 |
| 56 | Münster, Stubengasse | 445 | NW/DE | 5 | single | funerary | exterior | | regular | front | | North | 0 | male | adult | A | Winkler 2008 |
| 57 | Münster, Stubengasse | 446 | NW/DE | 5 | single | funerary | exterior | coffin | regular | front | extended | East | 0 | male | adult | A | Winkler 2008 |
| 58 | Müstair, Kloster St. Johann, Westhof | R762 | GR/CH | 5 | single | funerary | exterior | | disordered | front | other | West | | indet. | subadult | J | Hotz 2002 |
| 59 | Müstair, Kloster St. Johann, Westhof | W666 | GR/CH | 5 | single | funerary | exterior | | | | | West | 1 | female | adult | YA | Hotz 2002 |
| 60 | Nabburg, St. Maria | 478 | BY/DE | 3 | double | funerary | churchyard | | regular | front | extended | West | | male | adult | A | Hensch 2014 |
| 61 | Neubrandenburg, Ziegelbergstrasse | 99 | MV/DE | 3 | single | funerary | exterior | | disordered | front | extended | West | | male | adult | A | Prehn 2005 |
| 62 | Neukirchen, Friedhof St. Nikolaus | 26 | BY/DE | 3 | single | funerary | churchyard | | regular | back | | West | | indet. | adult | A | Ernst 1992 |
| 63 | Neukirchen, Friedhof St. Nikolaus | 29 | BY/DE | 3 | single | funerary | churchyard | | regular | back | | West | | indet. | adult | A | Ernst 1992 |
| 64 | Nördlingen, Spitalkirche | 190 | BY/DE | 2 | single | funerary | churchyard | | regular | front | | Southwest | 0 | female | adult | YA | Gebauer/Zäuner 2018 pers. comm. |
| 65 | Northeim, Grabkapelle | 2 | NI/DE | 1 | single | funerary | favored | | regular | side | | West | 1 | male | adult | A | Schütte 1989 |
| 66 | Potsdam, Nikolaikirche | 200 | BB/DE | 2 | single | funerary | churchyard | | | | flexed | West | 0 | indet. | adult | A | Jungklaus 2019 pers. comm. |
| 67 | Potsdam, Nikolaikirche | 266b | BB/DE | 2 | single | funerary | churchyard | | disordered | front | extended | West | 0 | indet. | adult | A | Jungklaus 2019 pers. comm. |
| 68 | Potsdam, Nikolaikirche | 278 | BB/DE | 2 | single | funerary | churchyard | | disordered | front | other | West | 0 | male | adult | A | Jungklaus 2019 pers. comm. |
| 69 | Prenzlau, Diesterwegstrasse | 308 | BB/DE | 5 | single | funerary | churchyard | coffin | | | | West | 0 | indet. | adult | A | Ungerath 2003 |
| 70 | Prenzlau, Diesterwegstrasse | 598 | BB/DE | 5 | single | funerary | churchyard | | | | extended | West | 0 | male | adult | A | Ungerath 2003 |
| 71 | Rieneck, Kloster Elisabethenzell | 7a | BY/DE | 3 | single | funerary | favored | | | | extended | West | 0 | male | adult | MA | Alterauge 2014 |
| 72 | Romont, Couvent Fille Dieu | 113 | FR/CH | 1 | triple | funerary | favored | | regular | front | extended | West | 0 | indet. | subadult | I | Bujard 2018 |
| 73 | Romont, Couvent Fille Dieu | 115 | FR/CH | 1 | triple | funerary | favored | | regular | | extended | West | 0 | male | adult | A | Bujard 2018 |
| 74 | Schloss Horst, Vorburg | 1296 | NW/DE | 3 | single | funerary | churchyard | | | | | North | 0 | male | subadult | J | Wiedmann 2010 |
| 75 | Schloss Horst, Vorburg | 1986 | NW/DE | 3 | single | funerary | churchyard | coffin | regular | front | extended | East | 0 | male | adult | A | Wiedmann 2010 |
| 76 | Schüpfen, Dorfstrasse 13 | 229 | BE/CH | 5 | single | funerary | exterior | | regular | front | extended | West | 1 | male | adult | MA | Alterauge et al. 2017 |
| 77 | Schweinfurt, Zeughausplatz | 268 | BY/DE | 3 | single | funerary | churchyard | | | | | Northeast | | male | adult | YA | Staskiewicz 2018 pers. comm. |

(Continued)

**Table 1.** (Continued)

| No. | Site | Grave | State/Country | Period | Burial type | Burial place | Burial location | Burial container | Arm position | Arm location | Leg position | Orientation | Grave goods | Sex | Age1 | Age2 | Reference (S1 File) |
|---|---|---|---|---|---|---|---|---|---|---|---|---|---|---|---|---|---|
| 78 | Schwyz, St. Martin | 314 | SZ/CH | 5 | single | funerary | churchyard | coffin | regular | side | extended | North | 0 | female | adult | OA | Cueni 2017 pers. comm. |
| 79 | Steinhausen, Pfarrkirche St. Matthias | 9 | ZG/CH | 5 | single | funerary | churchyard | | | | other | West | | indet. | subadult | J | Meyer/Doswald 2012 |
| 80 | Strausberg, Amtsgericht | 352 | BB/DE | 3 | single | funerary | churchyard | | | | | West | | indet. | adult | A | Wittkopp 2008, 2009 |
| 81 | Strausberg, Amtsgericht | 447 | BB/DE | 3 | single | funerary | churchyard | | | | | West | | male | adult | YA | Wittkopp 2008, 2009 |
| 82 | Tarrenz, Strader Wald | | 7/AT | 5 | single | non-funerary | exterior | | regular | front | flexed | North | 1 | female | adult | MA | Stadler 2013 |
| 83 | Templin, Kantstrasse 2 | 101 | BB/DE | 4 | triple | funerary | churchyard | | | | | West | | indet. | subadult | I | Jungklaus 2007 |
| 84 | Templin, Puschkinstrasse | 140 | BB/DE | 3 | single | non-funerary | settlement | | disordered | front | flexed | West | | female | adult | YA | Jungklaus 2018 pers. comm. |
| 85 | Ubstadt, Weiherer Strasse | 813 | BW/DE | 1 | single | non-funerary | settlement | | disordered | side | extended | East | | male | adult | A | Lutz 1997 |
| 86 | Unterseen, Kirche | 58 | BE/CH | 2 | single | funerary | favored | shroud | regular | front | other | Northwest | 0 | female | adult | OA | Eggenberger/Ulrich-Bochsler 2001 |
| 87 | Vérolliez, Chapelle des Martyrs | 16 | VS/CH | 5 | single | funerary | exterior | | disordered | side | flexed | West | 0 | indet. | adult | A | Auberson et al. 1997 |
| 88 | Vöhingen, Wüstung, Friedhof | 1702 | BW/DE | 2 | single | funerary | churchyard | | disordered | | other | Southwest | 0 | male | adult | YA | Arnold 1998 |
| 89 | Warburg, Hüffertstr. 50 | 750 | NW/DE | 3 | single | funerary | churchyard | | | | | Southwest | 0 | indet. | adult | YA | Bulla et al. 2013 |
| 90 | Wien, Minoritenkirche | 1/86 | 9/AT | 3 | single | funerary | churchyard | coffin | disordered | back | extended | East | 0 | male | adult | MA | Prohaska 2003 |
| 91 | Wilhemshof, Kloster Grobe | 59 | MV/DE | 3 | single | funerary | churchyard | | regular | front | extended | West | 0 | male | adult | MA | Jungklaus 2017; Biermann et al. 2017 |
| 92 | Winterthur, Stadtkirche, Westfriedhof | 13 | ZH/CH | 3 | single | funerary | churchyard | | regular | side | extended | West | | female | adult | OA | Jäggi et al. 1993 |
| 93 | Worms, St. Paul | 18 | RP/DE | 5 | single | funerary | churchyard | coffin | | | extended | West | 0 | indet. | adult | A | Grünewald 2001 |
| 94 | Worms, St. Paul | 29 | RP/DE | 5 | single | funerary | churchyard | coffin | disordered | front | | West | 0 | male | adult | MA | Grünewald 2001 |
| 95 | Zürich, Fraumünster | 4 | ZH/CH | 4 | single | funerary | churchyard | shroud | regular | front | extended | West | 0 | male | adult | YA | Moser et al. 2015 |

The information corresponds to the expression of variables used for the statistical analysis with additional information on the age categories. Empty cells represent missing information. States, cantons and countries are listed according to the ISO 3166–2 abbreviations. Grave goods: presence = 1, absence = 0, nd = indetermined, I = infant (0–12 years), J = juvenile (13–19 years), A = adult (above 20 years), YA = young adult (20–39 years), MA = middle adult (40–59 years), OA = old adult (above 60 years).

**Table 2. Definition of variables used in this study.**

| Variable | Expression | Definition |
|---|---|---|
| Period | 1 | High Middle Ages (10th-13th century AD) |
| | 2 | no differentation between period 1 and 3 (10th-16th century AD) possible |
| | 3 | Late Middle Ages (13th-16th century AD) |
| | 4 | no differentation between period 3 and 5 (13th-19th century AD) possible |
| | 5 | (Early) modern period (16th-19th century AD) |
| Burial type | single | |
| | double | at least one prone burial |
| | triple | |
| Burial place | funerary | specifically dedicated funerary place, e.g. churchyard |
| | non-funerary | non-funerary place, usually not used for burial |
| Burial location | churchyard | burial ground connected to a church |
| | favored | interior or prominent location to a church |
| | exterior | outside a church, burial ground |
| | settlement | habitation place |
| Orientation (of the head) | North | |
| | East | |
| | South | |
| | West | |
| | Deviations up to 45° | |
| Burial container | coffin | indicated by nails or wood remains |
| | shroud | stated by the excavator, indicated by pins or the tight position of the extremities |
| Arm position | regular | seen in "normal" medieval burials, e.g. arms on the chest, on the pelvis, both stretched out |
| | disordered | e.g. arms above the head, in extension from the body |
| Leg position | extended | |
| | flexed | |
| | other | e.g. tied, crossed or erected lower legs |
| Grave goods | 0 | absence |
| | 1 | presence, only deliberate furnishings (e.g. knives, coins, jewelry) |
| Sex | male | |
| | female | |
| | nd | indeterminate |
| Age | adult | ≥20 years |
| | subadult | <20 years |

centuries) and 27 (28.4%) to the early modern period (16th-19th centuries). Ten burials (10.5%) were attributed to period 2, and eight (8.4%) to period 4 (Table 4).

## Burial context and location

Prone inhumations are mostly single burials (82/95; 86.3%), nine derive from double burials (9.5%) and four from triple burials (4.2%). In the double burials, the following combinations of burial positions occur: 1) two prone burials on top of each other (e.g. Elten); 2) one

**Table 3. Radiocarbon dates of previously undated prone burials, measured at the laboratories of Bern (BE) and Zurich (ETH).**

| Site | Country | Grave nr. | Sample | Laboratory nr. | $^{14}$C age BP | ± 1σ | Cal 1σ (68.2%) | Cal 2σ (95.4%) |
|------|---------|-----------|--------|----------------|-----------------|------|----------------|----------------|
| Belfaux, Saint-Maurice | CH | 475 | MC I | BE-8255.1.1 | 310 | 20 | 1522–1642 AD | 1496–1646 AD |
| Bülach, Rathausgasse | CH | 86 | tooth | ETH-34325 | 360 | 50 | 1460–1630 AD | 1440–1640 AD |
| Büren, Chilchmatt | CH | 91 | MC II | BE-8939.1.1 | 1009 | 20 | 996–1030 AD | 986–1040 AD |
| Lausanne, Vidy | CH | 1558 | MC II | BE-8940-av | 402 | 20 | 1446–1480 AD | 1440–1616 AD |
| Nördlingen, Spitalkirche | DE | 190 | MC II | BE-9427.1.1 | 580 | 19 | 1320–1405 AD | 1310–1412 AD |
| Potsdam, Nikolaikirche | DE | 266b | humerus | BE-12804.1.2 | 626 | 23 | 1298–1390 AD | 1290–1398 AD |
| Unterseen, Kirche | CH | 58 | skull | BE-8766.1.1 | 689 | 21 | 1277–1297 AD | 1270–1384 AD |
| Winterthur, Stadtkirche | CH | 13 | MT I | BE-9383.1.1 | 407 | 19 | 1445–1474 AD | 1440–1612 AD |
| Zürich, Fraumünster | CH | 4 | skull | ETH-59666 | 363 | 27 | 1462–1620 AD | 1450–1634 AD |
| Zürich, Fraumünster | CH | 4 | tooth | ETH-59667 | 329 | 27 | 1498–1634 AD | 1481–1642 AD |

CH = Switzerland, DE = Germany, MC = metacarpal, MT = metatarsal, BP = before present.

individual in prone position and one in supine position on top of each other (e.g. Freiburg; Nabburg); 3) two prone individuals next to each other (e.g. Borkum). Within triple burials, the combination of prone and supine individuals in one grave is with three cases the most common, where either one (e.g. Templin) or two individuals (e.g. Romont) were buried in prone position.

Prone burials occur at a wide range of archaeological sites. 90 prone burials come from funerary contexts, mainly churchyards (n = 65/90, 72.2%). Several cases can occur at a single site, e.g. four (Hanau-Kesselstadt) or six prone burials (Flintsbach/Inn). Other funerary specimens have been uncovered in favored locations (n = 11/90, 12.2%), such as in the interior of a church (e.g. Altlichtenwarth) or chapel (e.g. Northeim). Regarding the chronological distribution per burial location, burials in favored location are predominant in the first period, while the later periods are dominated by churchyard (period 3) and exterior burials (period 5). Indeed, the correlation between burial location and period turned out significant (p < 0.001) (Table 4).

Another 14 funerary individuals in prone position (15.6%), all but one from the early modern period, were buried at a shared burial ground (e.g. Erding), outside the neighboring churchyard (e.g. Belfaux), outside the church walls (e.g. Vérolliez) or in an abandoned part of the cemetery (e.g. Schüpfen).

Five prone burials were found outside funerary contexts as isolated burials in settlements [68] or in the open landscape [69].

## Orientation, burial container, body position and grave goods

Information about the orientation was available for 94 graves, of which 61 (64.9%) showed a West-East or 13 (13.85%) an East-West orientation. Only seven individuals were clearly

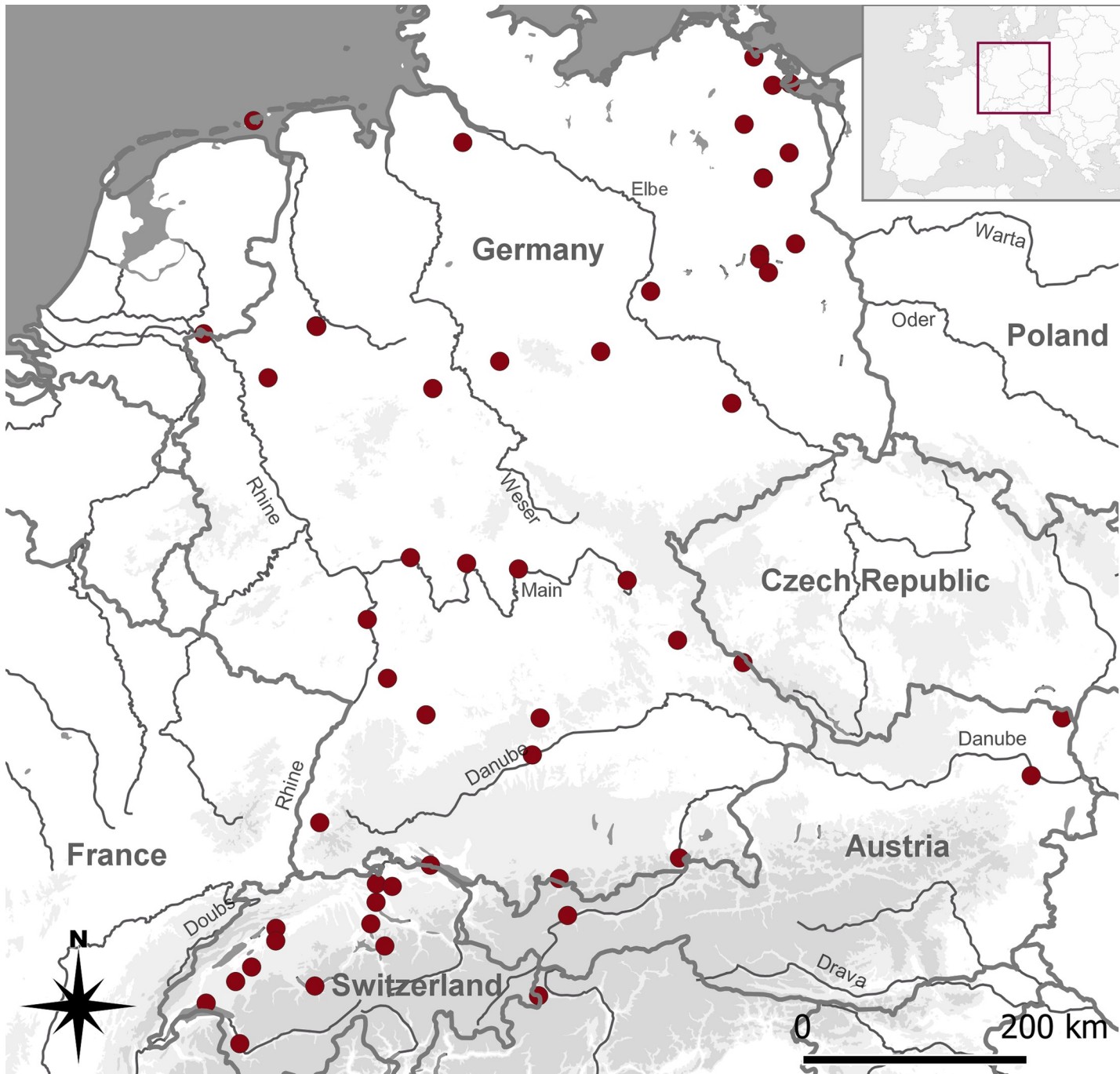

**Fig 1. Geographical distribution of medieval and post-medieval prone burials in Germany, Switzerland and Austria (n = 60).** One site might contain several burials in prone position. Complete dataset see Table 1. Basic vector map of Europe, reprinted from [67] under a CC BY license, with permission from Jonas von Felten, 2019.

oriented North-South (n = 5; 5.3%) or South-North (n = 2; 2.1%) while additional 13 graves (13.85%) were oriented with a deviation from North, South or West with up to 45˚. The orientation is significantly correlated with the period (p = 0.003) (Table 4), with West-East orientation being dominant throughout the different epochs but North-South and South-North orientation restricted to the late and post-medieval period.

**Table 4. Frequency of expressions of burial location, orientation, burial container and arm position per period, including the significance level (p, tested with Fisher's Exact Test).**

| | | Burial location n | | | | | Orientation n | | | | | | | | Burial container n | | | Arm position n | | |
|---|---|---|---|---|---|---|---|---|---|---|---|---|---|---|---|---|---|---|---|---|
| | | churchyard | favoured | exterior | settlement | | West | East | North | Northeast | Northwest | South | Southwest | | coffin | shroud | | disordered | regular | |
| Period | 1 (10th-13th ct.) | 10 | 8 | 0 | 1 | p<0.001 | 16 | 2 | 0 | 0 | 1 | 0 | 0 | p = 0.003 | 3 | 2 | p = 0.043 | 2 | 9 | p = 0.306 |
| | 2 (10th-16th ct.) | 9 | 1 | 0 | 0 | | 5 | 0 | 0 | 0 | 3 | 0 | 2 | | 0 | 1 | | 3 | 5 | |
| | 3 (13th-16th ct.) | 27 | 2 | 1 | 1 | | 23 | 3 | 1 | 1 | 0 | 0 | 2 | | 8 | 1 | | 3 | 13 | |
| | 4 (13th-19th ct.) | 6 | 0 | 1 | 1 | | 3 | 2 | 0 | 1 | 2 | 0 | 0 | | 1 | 1 | | 1 | 5 | |
| | 5 (16th-19th ct.) | 13 | 0 | 14 | 0 | | 14 | 6 | 4 | 0 | 0 | 2 | 1 | | 9 | 0 | | 9 | 10 | |
| | Total n | 65 | 11 | 16 | 3 | | 61 | 13 | 5 | 2 | 6 | 2 | 5 | | 21 | 5 | | 18 | 42 | |

Information about the burial container was available in 26 cases (n = 21 coffins, 80.8%; n = 5 shrouds, 19.2%). The association between burial container and period was significant (p = 0.043), due to the predominance of coffins in late and post-medieval times (Table 4).

In 60 cases, information on the arm position was available. While 42 individuals (70.0%) showed a regular position of the arms, 18 (30.0%) were observed with a disordered arm position. It is noteworthy that nine of these 18 individuals date to 16th to 19th century. However, the association between arm position and period is not significant (p = 0.306).

The leg position was considered in 56 cases, of which 44 (78.6%) are extended. Flexed legs and other leg positions (e.g. erected, crossed) were found in six cases (10.7%) each. The correlation between leg position and sex is not significant (p = 0.106), even though males predominantly had extended legs and females more frequently showed flexed legs.

For 59 burials, we had information about the presence or absence of grave goods. Only eight individuals (10.1%) were equipped with grave goods which included knives, coins and jewelry.

## Sex and age distribution

The majority of the 95 prone individuals were adult above 20 years (Fig 2). 84 individuals were identified as adults (88.4%), while in 39 cases a detailed anthropological age estimation was lacking so that they could only be described as adult *in sensu largo*. The detailed age estimation for the remaining 45 grown-ups revealed that 23 (51.1%) were young adults, 12 (26.7%) middle and ten (22.2%) older adults.

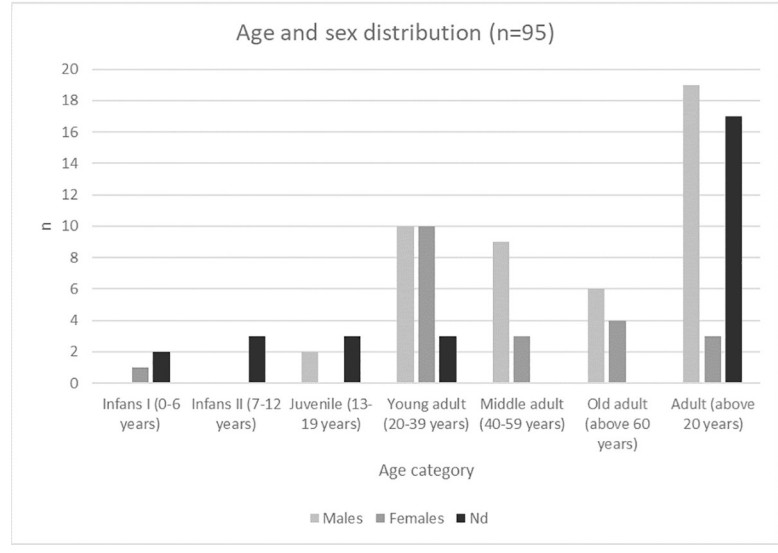

**Fig 2. Age and sex distribution of medieval and post-medieval prone burials (n = 95).**

Only 11 individuals (11.6%) are subadults. The three youngest individuals (0–4 years) were not buried alone, but in a double and triple burial with adult females. The juveniles, on the contrary, were buried in single graves. It is noteworthy that the association between burial type and age category is significant (p = 0.04).

Regarding sex, males represent the majority of individuals (n = 44/84, 52.4%), with both females (n = 20, 23.8%) and unsexed (n = 20, 23.8%) individuals relatively underrepresented (Fig 2).

## MCA

The eigenvalues of the first four dimensions obtained from MCA (accounting for 47.6% of the variance in the dataset) are as follows: 3.126 for dimension 1, 2.594 for dimension 2, 2.356 for dimension 3, and 2.169 for dimension 4. Due to marginal differences in variation, we are only presenting the first two dimensions (Table 5).

Fig 3A–3D visualizes the distribution of our data in the first two dimensions (accounting for the 26.6% of the total variance). The majority of burials cluster around the centroid, the latter representing the average distribution of all observations (in our case the 'average' prone burial), which features: single burial from funerary context, more specifically from a church-yard, regular arm position and extended legs.

Burials from non-funerary contexts are the most distant from centroid and cluster to the left of the bivariate space (Fig 3A). Their difference from the average is mainly due to the burial location and/or burial place but also due to irregular burial position (e.g. flexed legs). Burials in favored location form another relatively distinct group, at least the specimens that represent single burials of males dating to period 1. Accordingly, deviations from that pattern, e.g. Unterseen [70] or Elten [71], show the largest distance to the core group (Fig 3B). Other distinct groups include double or triple burials (Fig 3C) and subadult individuals (Fig 3D). All three groups show considerable overlaps in the plot (Fig 3C and 3D), revealing that subadult individuals often, but not exclusively, occur in double or triple burials.

**Table 5. Discrimination measures entered in the MCA.**

| Variable | MCA discrimination measures | | | |
|---|---|---|---|---|
| | Dimension | | | |
| | 1 | | 2 | |
| | Discrimination | Contribution (%) | Discrimination | Contribution (%) |
| Burial type | 0.220 | 7.02 | 0.234 | 9.02 |
| Burial place | **0.630** | 20.15 | 0.011 | 0.42 |
| Burial location | **0.841** | 26.91 | 0.031 | 1.20 |
| Orientation | 0.363 | 11.61 | 0.495 | 19.09 |
| Burial container | 0.032 | 1.02 | 0.223 | 8.62 |
| Arm position | 0.130 | 4.17 | 0.009 | 0.34 |
| Leg position | 0.233 | 7.47 | 0.446 | 17.20 |
| Period | 0.388 | 12.42 | **0.711** | 27.42 |
| Grave goods | 0.040 | 1.27 | 0.021 | 0.79 |
| Sex | 0.116 | 3.70 | 0.323 | 12.44 |
| Age category | 0.133 | 4.27 | 0.090 | 3.47 |
| **Active total** | 3.126 | 100.00 | 2.594 | 100.00 |
| **Inertia** | 0.284 | | 0.236 | |
| **% of variance** | 14.50 | | 12.10 | |

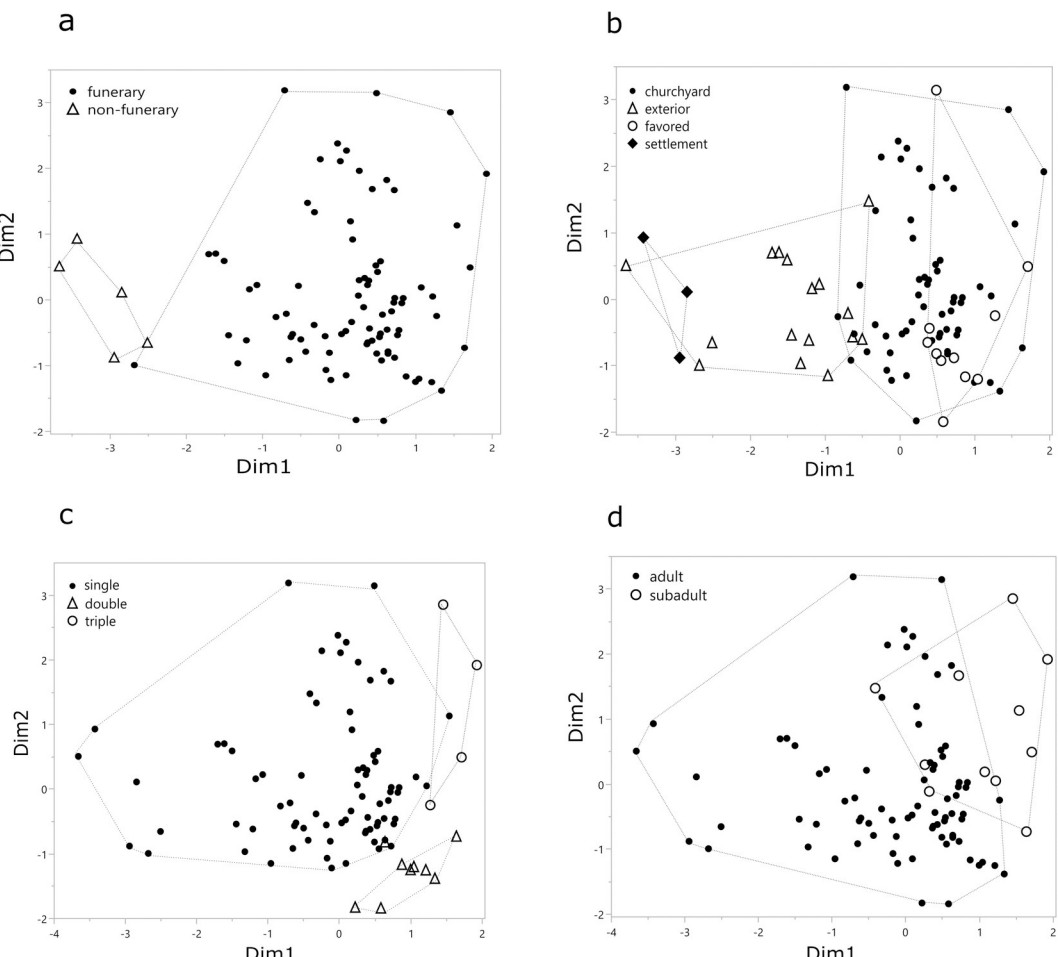

**Fig 3. Results of MCA: Distribution of individuals along the first and second dimensions.** The plot is repeated to highlight different variables. a) burial place; b) burial location; c) burial type; d) individuals age. The concentration of data around the centroid represents an average prone burial.

Discrimination measures for each variable were obtained (Table 5), and Fig 4 visualizes the correlation between variables and the principal dimensions of the MCA. There are some clear differentiating values allocated to each of the dimensions, above 0.5 respectively. The most discriminant variables for dimension 1 hierarchically are *burial location* and *burial place* (Table 5). Evidently, there is an important overlap between both variables since the category *funerary* is either associated with churchyard, favored or exterior burial while the non-funerary contexts represent either settlement or exterior burials. In this way, *burial place* and *location* also explain much of the data variability since correlations with other variables, such as *body position* and *orientation*, are rather high.

Regarding dimension 2, the most discriminant variable is *period*. It is also a relevant factor in the first dimension, and as previously mentioned the Fisher's Exact Test reveals a statistically significant association between *period*, *burial location*, and *burial orientation*. The variables *burial orientation*, *leg position*, *burial type* and *sex* present relevant and similar discrimination measures in both dimensions. The other factors cluster around the point of origin and reveal homogeneity thereof.

From data analysis, and its graphical representation, two MCA dimensions—termed *burial context* and *dating*—were identified. The factor *burial position* is superimposed by the dimensions mentioned beforehand.

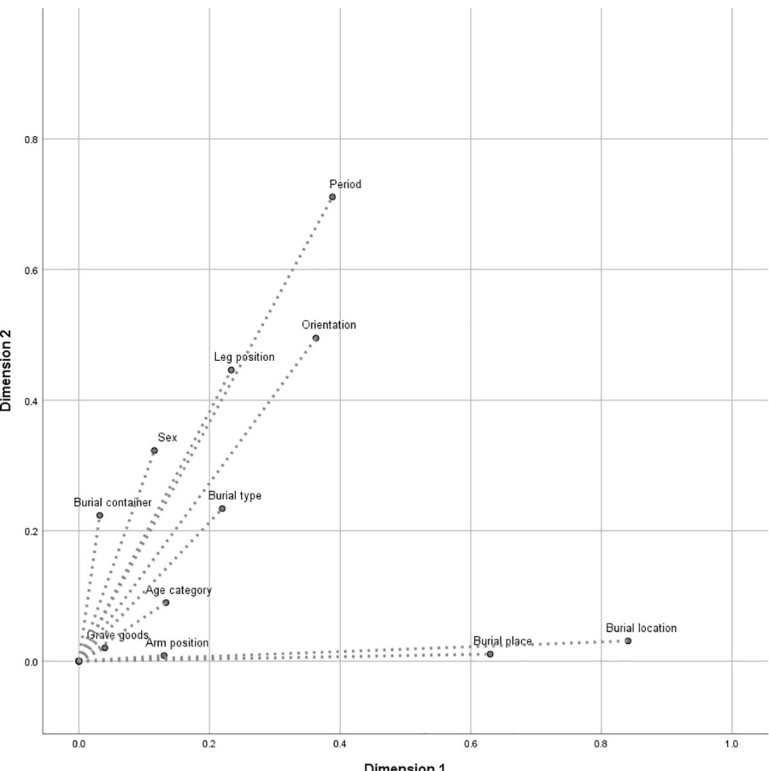

**Fig 4. MCA dimensions discrimination measures.** The variables burial place and burial location are correlated with dimension 1, and variable period is correlated with dimension 2. The variables orientation, leg position, burial type and sex show relevant discrimination measures with both dimensions.

## Discussion

### Limitations

Due to the inhomogeneous distribution of excavated areas and the possible lack of documentation or publication of findings, our dataset must be considered as an approximation of the real distribution of prone burials in the considered contexts. Various cases of prone burials not considered in our study are probably to be found in excavation reports housed in the archives of heritage state institutions, which makes this type of information not widely available. We are aware of further examples but the individual data on these burials were not (yet) accessible [57, 72]. Besides, we are aware that the recognition of prone position in the field is highly dependent on the excavation technique, the experience of the archaeologists and–ideally–the presence of a physical anthropologist on site, especially when it comes to densely occupied and disturbed medieval cemeteries. The high number of six prone burials from Flintsbach/Inn [73], for example, derives from a research excavation with special emphasis on the burial grounds, and some of these burials comprised only an arm or a foot in situ. These issues call for caution when interpreting our data. In any case, even considering this caveat, we believe that our study depicts interesting patterns.

The clustering in Northeastern Germany (Fig 1) is probably due to several factors, of which one, beside the activity of one of the authors as a field anthropologist, might be the increased construction activity during the last decades. Besides, the chronological focus, and therefore excavation and publication record, of a Heritage State Department can be an additional factor

in the availability of information on medieval and post-medieval graveyards. From an historical point of view, Slavic traditions might have still reigned in the regions east of the Elbe River [74]. Within a Christian framework, we can exclude religious denomination as a factor since there are examples from both Catholic (e.g. Worms) and Protestant (e.g. Hanau-Kesselstadt) sites from the Post-reformation period, suggesting that prone burials are rather a cultural than a religious phenomenon.

Isolated prone burials without context or grave goods are difficult to date and accidentally might be attributed to neighboring prehistoric sites. Concerning the individuals from churchyards, only systematic radiocarbon dating of the prone individuals and regular burials from the same site might reveal chronological gaps between those two. This is of particular importance since we assume for several individuals that they may have been buried next to (or within) a churchyard after its abandonment, possibly profiting from the vicinity of consecrated ground [75].

A major concern of the statistical analysis was that we are exclusively dealing with deviant prone burials, which tend to be somewhat similar and mainly consisted of churchyard burials. Therefore, the MCA was partly biased by the quantitative overrepresentation of such contexts. This effect was still increased with the replacement of missing values by the mode. The low amount of variance covered by the dimensions of the MCA reflects both phenomena. In the future, we would recommend an approach of comparing regular, deviant and execution site burials in order to test whether different burial categories cluster together. Being an enormous undertaking, a possible workaround would be to start with one or two sites from the same region, which contain all three burial categories.

## Prone burials as part of the norm

With the exception of their atypical position, most prone burials have an otherwise normal appearance. Their rarity suggests that we are dealing with personalized acts for specific individuals. Like other medieval or post-medieval graves, the majority of prone burials are single burials. However, face-down inhumations also occur in double burials, a pattern that has already been noticed for the Early Middle ages [50, 76]. The individuals are usually buried on top of each other while one individual is buried in prone and the other one in supine position. In this way, they either end up face-to-face or back-to-back, however, there are also cases in which the individuals are buried in opposite orientation. The position establishes a strong personal connection between the individuals who probably died at the same time (and possibly of the same cause). It is noteworthy that the few children in prone position all derive from double or triple burials [77] (Fig 3C and 3D). In this regard, those burials are very similar to regular multiple burials in supine position that often include both adult and sub-adult individuals [78]. The interpretation of such multiple burials in the same grave pit ranges from a familial relation between the deceased to more profane reasons, such as reduced burial fees, space-saving and pauper's graves [79]. At least for the children, it seems unlikely that the prone position was intended as a punishment due to their assumed innocence.

Regarding the burial container, both coffins and shrouds have been observed within our prone burials. In this regard, they differ from contemporaneous prone burials in Belgium which were exclusively inhumations in simple earth pits [80]. The orientation of our prone burials predominantly follows the standard West-East orientation of medieval burials whereas the few North-South oriented examples date to the modern period (18th/19th century) during which a North-South-orientation of graves generally becomes more frequent (Table 4).

Period- or confession specific grave goods are found occasionally but reflect furnishing or funeral customs of the time rather than the deliberate provision or denial of grave goods to the

deceased. Elements of clothing tell us whether the body was dressed at burial and sometimes even mark this person as foreigner, such as in Worms where a French soldier, as identified by his uniform, was buried face-down in the parish cemetery [81]. Grave goods, such as iron knives and purses (with coins), were sometimes found close to the body, like in Schüpfen, Switzerland (Fig 5) [82]. We think that relatives or undertakers refused to take them, possibly because the individual died of an infectious disease or was found in an advanced state of decomposition (e.g. after drowning, death outdoors). Knives, belt buckles and coins were not a frequent, but nevertheless regular, grave good among ordinary burials, too [83, 84].

The majority of our prone individuals were buried with extended legs and the arms to the front with hands on the chest (Fig 5), on the abdomen (Fig 6) or the pelvis or stretched out along the body. Comparative research studies from France [85], Denmark [86] and Switzerland [87] suggest that these positions follow the average patterns for their time in Central Europe. As expected, the position of the arms to the front is the most frequent in our sample, followed by the arms to the side. Arms positioned on the back may indicate some kind of fixation, although this hypothesis cannot be substantiated on the basis of the available archaeological data [88, 89]. Furthermore, we have to bear in mind that the recording of the detailed position is influenced by several factors: integrity of the body, excavation technique, and publication record.

The disordered arm position in a large number of our individuals suggests a certain degree of hastiness and carelessness during inhumation (Fig 7). While some burials look as if the body was thrown into the grave pit, other measures were meant for space-saving, e.g. erected [70] or tied legs [90]. Flexed legs may be seen in the same context but are so far restricted to female burials [91].

In some cases, atypical body positions have been interpreted as unintentional and possibly accidental [92]. Proposed explanations include mistakes of the undertakers [93] or a misplacement of the body wrapped in a shroud. Besides, live burial has been suggested as an explanation for prone position, in order to force open the coffin with the back against the lid when revived [89, 94, 95]. These explanations seem difficult to accept for the majority of our cases due to the rare, but diachronic occurrence of this burial type, but we cannot exclude them for individual cases.

Age-at-death distribution in our dataset shows a predominance of young adults among the individuals with specific age-at-death estimation (Fig 2) and a higher proportion of males. Admittedly, the high number of individuals classified as adults in *sensu largo* might bias this observation. Nonetheless has Gardeła [31] demonstrated the same tendencies for early medieval Poland (10th-13th centuries) but lacks to explain his observation. Concerning the age distribution, an analogue observation was made for burials with stoning, suggesting that juveniles and young adults were particularly at risk of being regarded as deviant because of their behaviour or circumstances of death [96]. Given the interpretations as criminals or suicides, modern data suggest indeed that young men are the group most prone to this kind of actions [97].

A marginal location of a prone burial within a churchyard is often interpreted as an additional sign of social deviancy. Depending on the extent of the excavation, our burials were recorded as having "the largest distance to the church" [98], as located at the churchyard's periphery (as defined by the churchyard wall) [75] (Fig 5) or as being at the least favorite site of the church. This assessment only works as long as the church serves as a reference and the proximity to it as a social indicator and might therefore change during the early modern period. A similar argument can also be applied for a diverging burial orientation, which may be due to evolving burial customs [99].

In addition, a few of our sites are not connected to a church but were rather established as burial places for a specific social group, often for poor, sick or hospitalised individuals [13], or during times of war or epidemics [100]. Those groups were not only separated from the population during life, but also after death as an ultimate exclusion mechanism. For instance, the cemetery at Münster

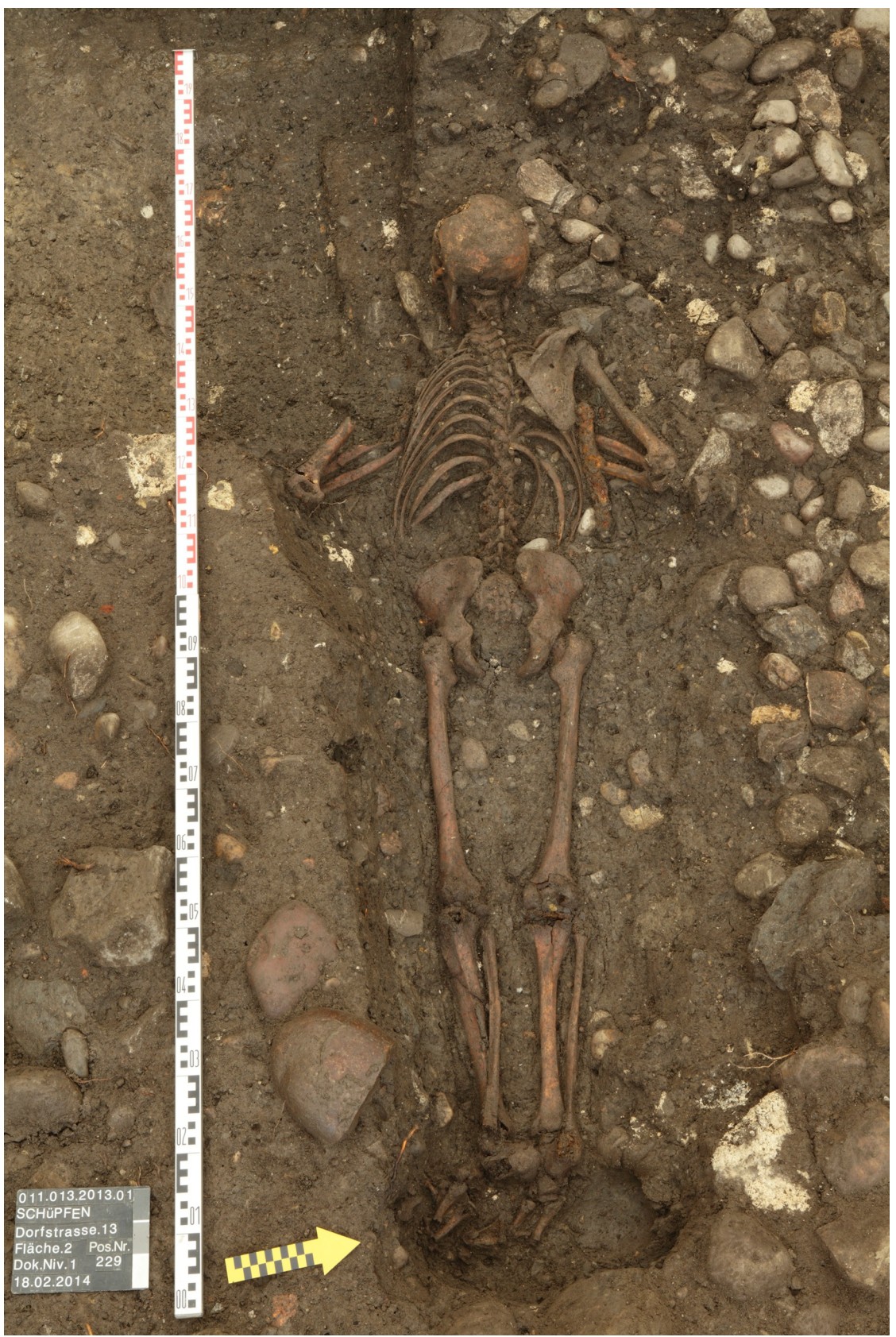

**Fig 5. Prone burial from the churchyard of Schüpfen (CH), grave 229.** The male individual is equipped with grave goods (knife, purse) in the crook of the arm. Note the careful arrangement of the limbs and the West-East orientation of the grave. The burial is located in an abandoned part of the churchyard, but inside the cemetery wall. (© Archäologischer Dienst des Kantons Bern, Daniel Breu).

Stubengasse was reserved for the patients of the Clemens hospital who could not afford a proper burial [101]. Irregular burial positions, more frequent than at other sites, evoke the indifference and disrespect towards their corpses [95]. In addition, medico-anatomical interventions, e.g. craniotomy, are reported from the same cemetery, and a headless prone individual in Greifswald has been associated with autopsy of delinquents for the purpose of anatomical training [102].

All in all, the majority of our prone churchyard burials do not contravene normative funerary provisions. As in Viking age Sweden [36, 37] and medieval Finland [34], the bodies seem to have been cared for, and the graves had been prepared and furnished according to the general customs of the community. Notably, some exceptions suggest a rather hasty and careless funerary procedure. Furthermore, the prone burials from churchyards are missing the factors that are usually associated with deviant burials, such as decapitation, stoning, or nailing [4, 28, 48, 96]. This suggests that prone inhumation, at least for the geochronological context under study, did not necessarily represent an exclusionary act against the deceased.

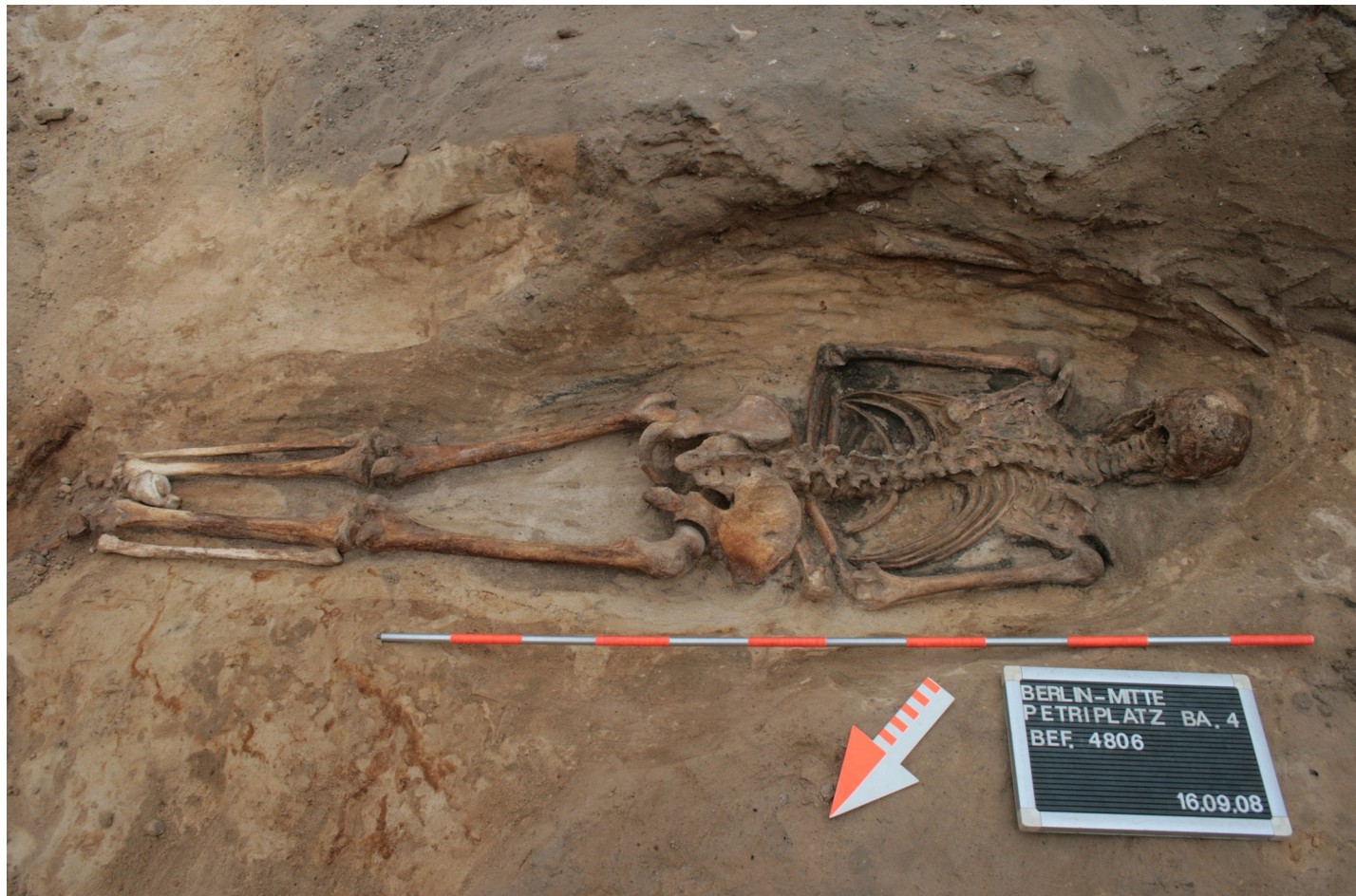

**Fig 6. Prone burial from the churchyard of Berlin Petriplatz (D), grave 4806.** The male individual was carefully placed in the pit wrapped in a shroud (© Landesdenkmalamt Berlin, Claudia Maria Melisch).

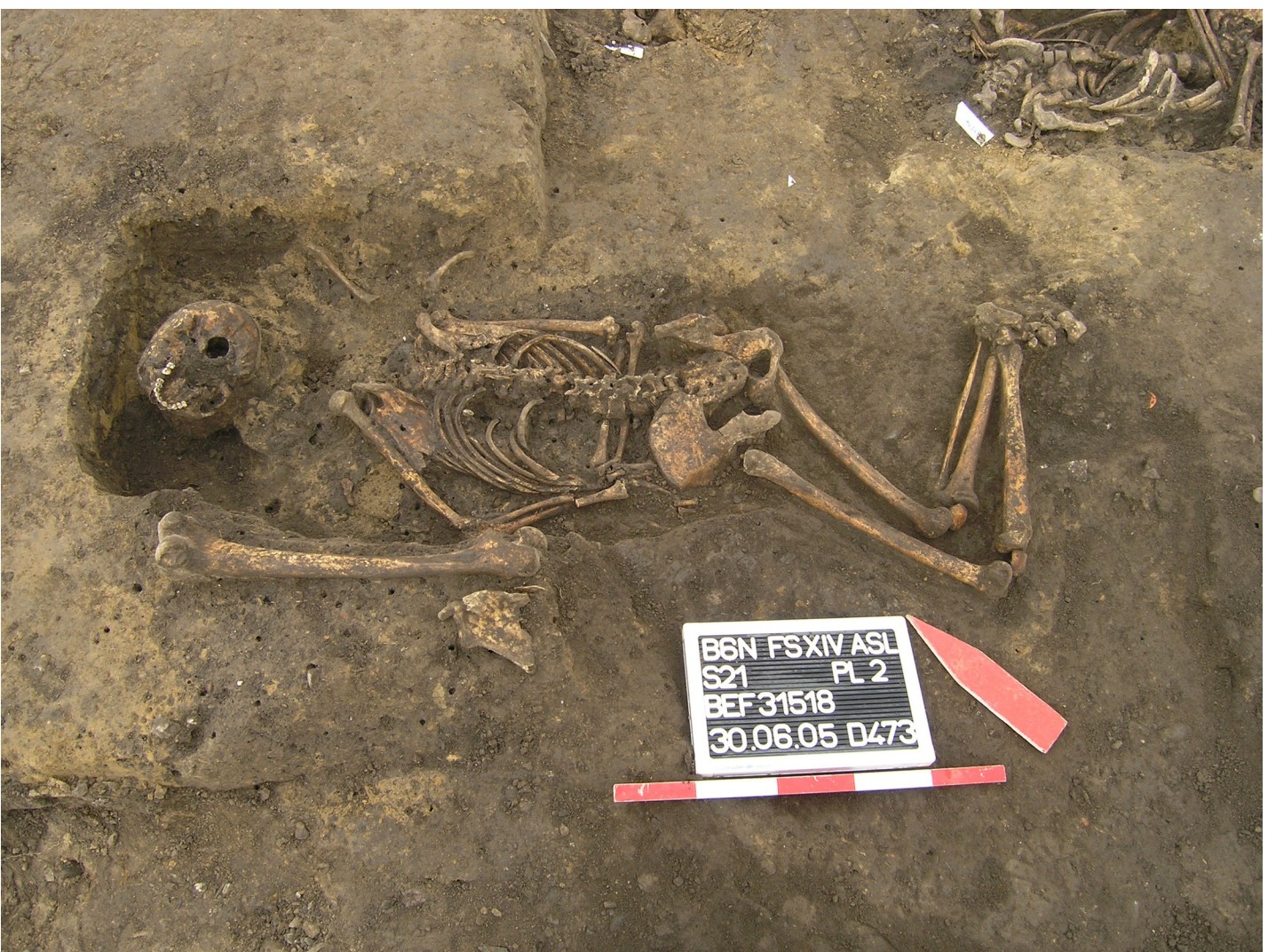

**Fig 7. Prone burial from the churchyard of Klein Hoym (D), grave 31518.** Despite the disordered position with flexed legs the female burial is well-integrated into the graveyard. Note the secondary dislocation of the skull. (© Landesamt für Denkmalpflege und Archäologie Sachsen-Anhalt, Andreas Selent).

## Prone position as a sign of *humilitas*

Prone position has also been suggested as a sign of *humilitas*, the Christian virtue of being humble and devoted to God. It would recall a gesture of submission as during proskynesis, priestly ordination or of penitents awaiting their resumption into church [103]. This interpretation historically refers to one specific incident, namely the burial of Pepin the Short in Saint-Denis close to Paris in 768 AD. On the occasion of the reopening of Pepin's grave in 1137, abbot Suger reported that Pepin, son of Charles Martel and the first Carolingian to become king of the Franks, chose to be buried face-down in front of the church front portal as a sign of humbleness and in expiation of his father's sins [47, 104]. The narrative suggests that being buried prone was considered as an expression of devotion, humility and penitence in the 12th century.

The present archaeological evidence, although scarce, might support this hypothesis. The MCA has revealed a meaningful association between burial location, period, and sex. There are a few prone burials from favored funerary locations, such as the interior of a church [105],

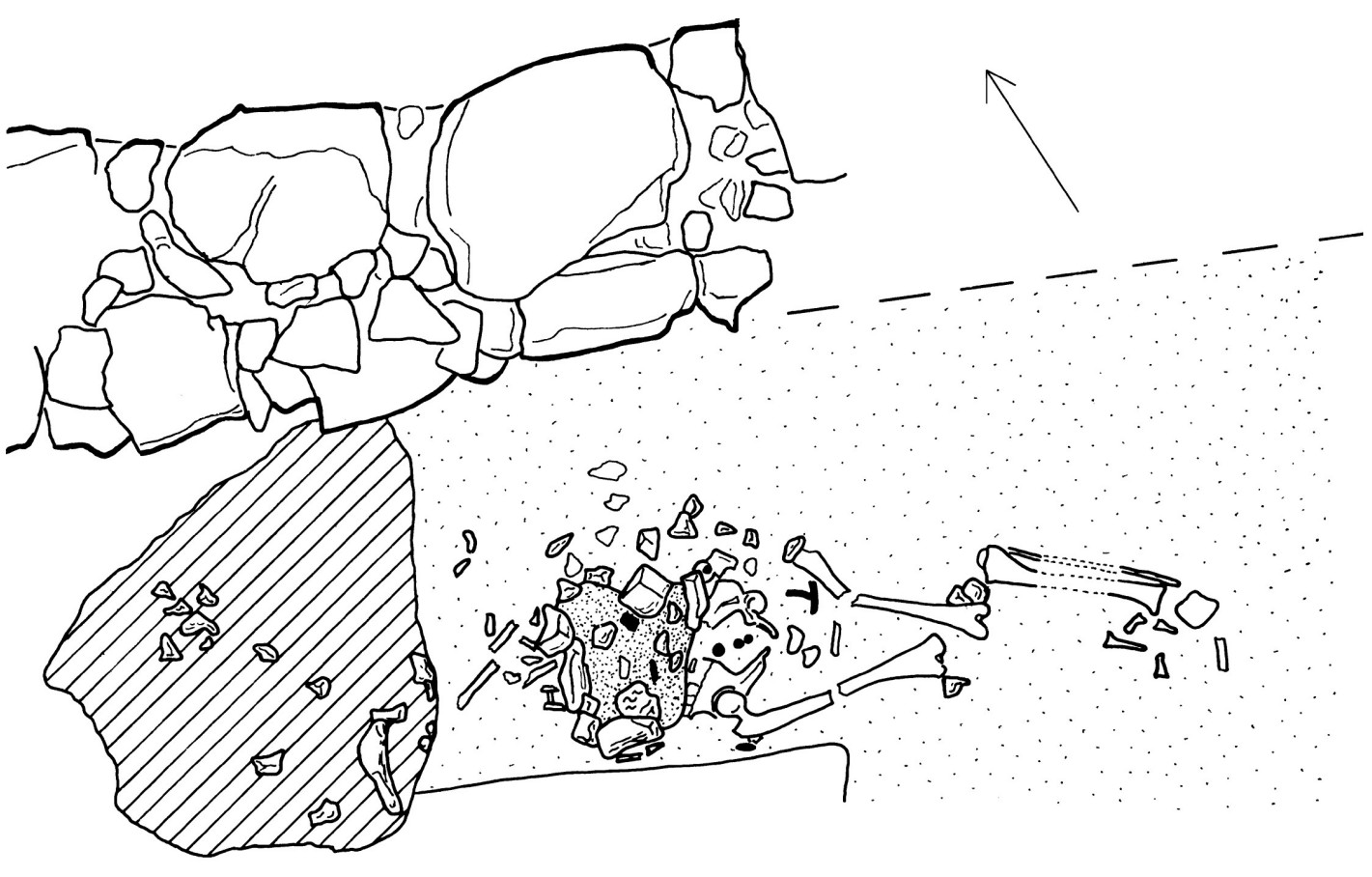

**Fig 8. Prone burial close to the portal of St Peter's church on the Kleiner Madron near Flintsbach/Inn (D), grave 230/519.** The man was equipped with four coins, polished stones and a Mithraic gem. (© Thomas Meier).

in front of a church gate [73] (Fig 8) or from a chapel [106]. A burial in close proximity to the altar (*ad sanctos*) was a privilege of the clergy, the nobility or patricians and promised salvation due to the immediate presence of God. Indeed, some deceased were assumed to originate from a clerical [107] or noble context [71, 73, 108], underlined by clothing and burial goods. In this context, prone position was nearly exclusive to high medieval male individuals. The burials of lay persons from Strausberg [109, 110] and Neukirchen [88] are also interpreted in this manner, extending this practice even to the Late Middle Ages.

Parallels are to be found in Alsace: several burials of a 13th century Dominican convent from Guebwiller, where the bodies were placed in prone position and North-South-orientation, were expressing humbleness in the spirit of Saint Dominicus [111]. An early dissemination of the idea has recently been suggested for the prone burials from Viking age Sweden [36, 37].

## Prone burials as social exclusion

Five individuals originate from non-funerary contexts; in all these cases, no other burials were observed in the surrounding area and the next churchyard was several hundred meters away. However, traces of a former or contemporary settlement were often found with those burials. In the lack of associated objects, the dating of these burials is often challenging and is mostly done via the stratigraphy of the surrounding structures. The majority of our non-funerary

burials date to late or post-medieval times, with the exception of the settlement burial from Ubstadt [112]. The deceased were buried in simple pits without coffin or shroud. The attested burial positions are characterised by a large variation with both irregular and regular arm and leg positions. The burials did not contain grave goods or clothing elements, except the female from Tarrenz, which Stadler considers as a healer, sorceress or sutler [69]. Their burial context evokes a disposal of the bodies along animal carcasses, for example at a knacker's yard [52], or in a settlement [68, 112, 113], while the careful positioning of the extremities manifests at least a certain degree of attentiveness of the burying community. In the case of Ubstadt, the burial location inside a settlement may also continue early medieval traditions to bury some persons at the fence or under the gutter of farmsteads [114].

In accordance with Carelli [115], Gordon [39] and Sörries [25], we suggest that that motives of people burying bodies outside consecrated ground fall into two categories: either individual motives of purely personal character and related to the manner of death or public motives represented by acts committed by society in the form of execution and burial. Suicide and homicide, and accordingly the attempt to hide a dead body in secret, fall into the first category. Execution, on the other hand, represented a public act. From the religious point of view, the corpses of delinquents could have been interred in the churchyard's consecrated ground, since punishment for the offence had been carried out. Except for upper class execution victims who were granted a burial in the churchyard, the practice, however, was often different since death was regarded as an insufficient punishment [116]. Accordingly, the corpses were buried below or around the gallows in irregular positions without care [60–62]. Interventions comprised non-traditional positioning, burdening, fixation of the extremities, or violating the bodies' physical integrity [117]. Apart from supine graves, prone or side positions as well as partial inhumations among animal carcasses occur at execution sites [60–62]. In this regard, our prone burials from non-funerary contexts exhibit strong similarities to the execution burials, even though they do not show evident traces of violence on the skeleton. Hence, they could be witnesses of the judicial and social demarcation practices of the Middle Ages and early modern period.

In addition, other marginal groups have been compelled to use execution sites for their burials. The burial ground of Erding in the vicinity of the gallows was probably used by non-local travelling clans, possibly gypsies [118, 119]. The site included primary and secondary burials in extended supine, prone and side positions as well as offering pits. Among these four prone burials, we highlight the inhumation of a pregnant woman who may have been seen as a potential revenant [120].

At some places, separate pauper's graveyards were established for the outlaws and poor [53]. Immoral lifestyle, involvement in witchcraft and sorcery, heresy, mental or physical disabilities and foreignness have stigmatized individuals as social outcasts [121], possibly leading to a burial outside the churchyard [122]. Additionally, suicides and victims of accidents or homicides fell into the same category. If there was no execution site or pauper's cemetery at hand, they were reported to be buried near the crossroads, in private gardens or on the boundaries of various territories or fields [100, 116, 39]. The spatial isolation was intended as a post-mortem exclusion and humiliation.

Due to the remoteness of the places, this category is currently strongly underrepresented in the archaeological record and thus among our sample.

### Prone position against the revenant dead?

As we have shown above, favored burial location and funerary equipment have led to rather positive interpretations of high medieval prone burials, while late and post-medieval

specimens are rather interpreted as deviant. Although not exclusive, these perspectives reflect diverging research traditions in medieval archaeology, with the current domination of sensationalist views [55]. Notably, the awareness of and interest in deviant burials has increased over the past 30 years, as did the general knowledge of medieval burial practices [19]. But, we are still suffering from a geographically imbalanced state-of-research. In Western Germany, Switzerland and Austria the state-of-research is in favor of the church interior rather than the surrounding cemeteries [123], resulting in an over-representation of high-status burials. This focus on the church and high-status burials does, however, not explain why our results suggest that prone burials were more widespread during the late Middle Ages than before, nor why they became more frequent outside churchyards in early modern times. In Eastern Germany, on the other hand, research has focused on the "feudalist period" for a long time, while in recent years many late and post-medieval cemeteries have been excavated during construction activities.

So far, we find the greatest number of prone burials in the former Slavic territories of Brandenburg and Mecklenburg-West Pomerania, dating to the time after the full establishment of Christianity in the 12th/13th century. Apart from prone and side position, other apotropaic practices are associated with Slavic traditions, too. Body manipulation, decapitation, stoning, nailing, sickles across the throat and stakes through the heart have been observed in Eastern Germany [96, 124], Poland [48, 125, 126], Czech Republic [30, 33], and Slovakia [127, 128]. The climax of these practices is from the 11th to 13th century, after the transition from cremation to inhumation burials and the introduction of Christianity. They continue to occur, but less frequently, in later cemeteries. Moreover, the combined practices also occur outside or at the periphery of the Slavic region, namely in Bavaria [129–131].

The described actions affected the corpse itself, either to hold it in the grave (e.g. stoning, nailing, prone position) or to banish the person for good (e.g. decapitation). Evidently, the underlying perception was that the person was undead and capable of doing harm to the living, a notion which can first be traced in the early and high medieval Slavic Balkans. Western European revenants of that time were returning for more friendly purposes to warn and admonish their relatives and friends of the times to come [46]. Such ghost stories served to educate Christians about the doctrine of the purgatory and to convince them of the efficacy of suffrages for the dead [46, 132]. It was only in the course of the Late Middle Ages that the Eastern European belief in the undead spread to Western Central Europe as well. Since then, two main categories of revenant dead appeared. European ethnology classifies two chronologically and regionally diverse perceptions [133, 134]: The *Wiedergänger* are believed to physically return to the world of the living, either to avenge some experienced injustice, or because their soul is not ready to be released, due to their former way of life. Their time as revenants on earth may be limited, and after the punishment they can achieve salvation. In other cases, the revenants are condemned to eternal damnation; the living have to apply repelling and banishing measures to the corpse [133]. *Nachzehrer*, on the contrary, are assumed to be deceased which stay in their graves and harm the living from there. They usually originate from an unusual death such as suicide or accident. Their main goal is to drain vital force from their relatives. The *Nachzehrer* devour their own bodies, including their funeral shrouds, and in doing so, cause smacking sounds. They are also associated with epidemic sickness; whenever a group dies from the same disease, the person who dies first is labelled to be the cause of the group's death [133]. The transformation to both *Wiedergänger* and *Nachzehrer* happens after death without external stimulus and the state is not communicable to the living. It was only throughout the 18th century that reports on vampire attacks became a clear element of European folklore, even though the incidents were limited to Serbia, Romania, Poland, Lithuania and Russia [135]. During this period, the perception of vampirism as being a communicable state evolved

while the modern Vampire perceptions were shaped by 19[th] century English literature tales [136].

The extent of the preventive measures might thus reveal whether the contemporaries feared the corpse to walk around or to act from the grave, but it is also quite possible that measures were decided based upon actual needs. Whatever reasons were keeping the deceased on earth— premature death, anomalous lifestyle or punishment for committed sins -, they were obviously not severe enough to deny the body a churchyard burial. However, we need to keep in mind that the transformation of the deceased into a revenant dead might not necessarily be evident during the funeral, but could also happen later, by revelation to his relatives through dreams or harmful actions. In these cases, the graves must have been reopened later and the bodies turned over, decapitated or manipulated in another way. However, we have no indications so far for secondary burial openings of prone graves so that we believe that the dead were buried prone from the outset. As a limitation, archaeologists are mostly not capable to distinguish between practices occurring during burial or within the first months after. In few cases, later burial openings and secondary manipulations that might represent belated practices against a supposedly undead, for instance covering the grave with stones, could be detected [137, 138].

The rapid spread of epidemic diseases in the Late Middle Ages, namely plague, and later also of typhoid fever, syphilis and cholera, promoted the fear of the dead, not only in the sense that people were afraid of infection, but also because of an intensified dealing with corpses. The perception of reanimated corpses was surely influenced by the experience of decomposing, moving and smacking bodies. The fact that prone position is lacking from attested, plague row burial sites (mass graves not included) could be indicative of prone burials dating to the early or late phases of the epidemic [139] during which otherwise normal burial practices were kept but the disease was feared the most. Prone position could therefore represent an act to protect the living by restraining the dangerous dead from returning and the disease from spreading [43].

Schürmann [140] has argued that the fear of *Nachzehrer* has spread from Silesia to Central Germany following epidemics and has reached the Rhine through Thuringia and Hessen during the 16[th] century. To him, the relative uniformity of the beliefs around the *Nachzehrer*, especially their smacking sound, reveals their recent introduction into German folklore. The observed chronological and geographical distribution of prone burials agrees very well with this observation.

Besides, to the best of the authors' knowledge, no such practices like stoning, nailing and decapitation have been noticed so far in medieval churchyards in Western Germany or Switzerland. The very few exceptions include secondary manipulations of the grave and do not fall into the period under study [137]. Thus, those practices seem to be limited to Eastern Europe and the former Slavic territory [48].

The lack of evidence in Western Central Europe might suggest that:

- the effectiveness of these practices was doubted;

- these rites were not part of the burial repertoire of those regions;

- the belief in dangerous revenants did not exist outside the Slavic area [46].

Prone position, on the other hand, has ever existed in Western and Central Europe [28, 35] while other apotropaic practices did not. It is a reasonable inference that the belief in the undead did not exist in the west until the end of the Middle Ages. We therefore hypothesize that the spread of infectious diseases, especially plague, in late medieval times was an important stimulus for the introduction of the belief in the dangerous dead. However, judging from the scarce evidence, the idea did not fall on fertile ground everywhere.

## Conclusion

With this study, we are only beginning to embrace the multiplicity of meanings of burying people face-down in the Middle Ages and early modern period. Simplistic interpretations can neither be maintained in regard of the chronology nor of the typology of the graves. Clearly, prone burial was applied across the spectrum of sexes, age, and wealth and it is likely that the rite had different motivations, especially when differentiating between funerary and non-funerary contexts. Prone burials appear as conscious and efficacious acts that occurred in parallel to the normative burials at the churchyards, representing *humilitas* during the High Middle Ages and exclusionary or protective measures against dangerous dead in later periods. In non-funerary contexts, the disposal and postmortem humiliation of the deceased was probably the motivation for face-down position.

We therefore plead for an analysis on individual basis, stressing the necessity for more detailed cemetery and regional studies. This would allow a more contextual approach, which takes archaeological context and pattern as its starting point, but also requires the collaboration with other disciplines, such as history, ethnology, physical anthropology, studies of religions. Further investigations of deviant burials hold the potential for nuancing our understanding of the medieval world and the mentalities of its inhabitants.

## Supporting information

**S1 File. Reference list to Table 1.**
(DOCX)

## Acknowledgments

We would like to thank all colleagues and state heritage departments who have provided us with unpublished information on prone burials in their respective research area. In addition, we thank Christine Cooper for language editing.

## Author Contributions

**Conceptualization:** Amelie Alterauge.

**Data curation:** Amelie Alterauge, Thomas Meier, Bettina Jungklaus.

**Investigation:** Amelie Alterauge, Bettina Jungklaus.

**Methodology:** Marco Milella.

**Project administration:** Amelie Alterauge, Sandra Lösch.

**Resources:** Sandra Lösch.

**Software:** Marco Milella.

**Supervision:** Sandra Lösch.

**Validation:** Thomas Meier, Marco Milella.

**Visualization:** Amelie Alterauge, Marco Milella.

**Writing – original draft:** Amelie Alterauge, Marco Milella.

**Writing – review & editing:** Amelie Alterauge, Thomas Meier, Bettina Jungklaus, Marco Milella, Sandra Lösch.

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
