## [Decision Letter · Decision Letter 0]

16 Jun 2020

PONE-D-20-13088

Between belief and fear - Reinterpreting prone burials during the Middle Ages and early modern period in German-speaking Europe

PLOS ONE

Dear Dr. Lösch,

Thank you for submitting your manuscript to PLOS ONE. After careful consideration, we feel that it has merit but does not fully meet PLOS ONE’s publication criteria as it currently stands. Therefore, we invite you to submit a revised version of the manuscript that addresses the points raised during the review process.

All comments must be addressed and language editing needs to be done by a native speaker before re-submission.

We look forward to receiving your revised manuscript.

Kind regards,

Peter F. Biehl, PhD

Academic Editor

PLOS ONE

Journal Requirements:

2. In your manuscript, please provide additional information regarding the specimens used in your study. Ensure that you have reported specimen numbers and complete repository information, including museum name and geographic location.

For more information on PLOS ONE's requirements for paleontology and archaeology research, see https://journals.plos.org/plosone/s/submission-guidelines#loc-paleontology-and-archaeology-research.

3. Please include your tables as part of your main manuscript and remove the individual files. Please note that supplementary tables should remain as separate "supporting information" files.

5. We note that Figure 1 in your submission contain map images which may be copyrighted. All PLOS content is published under the Creative Commons Attribution License (CC BY 4.0), which means that the manuscript, images, and Supporting Information files will be freely available online, and any third party is permitted to access, download, copy, distribute, and use these materials in any way, even commercially, with proper attribution. For these reasons, we cannot publish previously copyrighted maps or satellite images created using proprietary data, such as Google software (Google Maps, Street View, and Earth). For more information, see our copyright guidelines: http://journals.plos.org/plosone/s/licenses-and-copyright.

You may seek permission from the original copyright holder of Figure 1 to publish the content specifically under the CC BY 4.0 license. 

If you are unable to obtain permission from the original copyright holder to publish these figures under the CC BY 4.0 license or if the copyright holder’s requirements are incompatible with the CC BY 4.0 license, please either i) remove the figure or ii) supply a replacement figure that complies with the CC BY 4.0 license. Please check copyright information on all replacement figures and update the figure caption with source information. If applicable, please specify in the figure caption text when a figure is similar but not identical to the original image and is therefore for illustrative purposes only.

Additional Editor Comments (if provided):

Your manuscript has now been seen by two referees, whose comments are appended below. You will see from these comments that while the referees find your work of great interest, they have raised some concerns that must be addressed before re-submission. Most importantly, the manuscript has to be language edited by a native speaker.

Reviewers' comments:

Reviewer's Responses to Questions

**Comments to the Author**

1. Is the manuscript technically sound, and do the data support the conclusions?

Reviewer #1: Yes

Reviewer #2: Yes

2. Has the statistical analysis been performed appropriately and rigorously? 

Reviewer #1: Yes

Reviewer #2: Yes

3. Have the authors made all data underlying the findings in their manuscript fully available?

Reviewer #1: Yes

Reviewer #2: Yes

4. Is the manuscript presented in an intelligible fashion and written in standard English?

Reviewer #1: Yes

Reviewer #2: No

5. Review Comments to the Author

Reviewer #1: I am convinced by the data set as well by their interpretation. Especially the belanced argument for different contexts seems to be plausible and shows indirectly, that archaeology produces sometimes containers which incorporate very different features. It becomes apparent that there is no single explanation any more. Furthermore, prone burials can be seen as "part of the norm" - they do not necessarily represent deviant burials in general.

Just some remarks:

Fig. 2 does not necessarily show young adults as the main group, as the older adults and the especially the individuals just classified as "adult" may indicate the presence of all age groups.

Looking for the burial places, it is apparent that more than two thirds have been buried within the churchyard, which means no exclusion!

The climax of prone burials during the period of the 11th to 13th century (p. 12) may be due to research activities. Cemeteries of this "pre-Christian" time has been excavated much often than later burial places. And this is also the case in comparison with the West - which may lead to an over-estimation for the East.

"Wiederkehrer" and "Nachzehrer" are terms of the modern "Volkskunde" I suppose, and the data go back to the Handatlas der deutschen Volkskunde. Therefore we should be careful with their interpretation, especially the Handwörterbuch des deutschen Aberglaubens is problematic, as it has been only reprinted in 1987 as cited here, but originally printed in the 1920s. Nevertheless, the argument of an increase of dangerous dead seems to be plausible.

All references are listed in order of their occurence - could they be ordered alphabetically?

Reviewer #2: This is an important survey and I thoroughly enjoyed reading this paper and reflecting on its results. As the authors clearly state, whilst the topic has been intensively studied in Britain and increasingly in Poland and parts of Scandinavia, a survey of Central Europe (even if focusing largely on German-speaking areas) is very much needed. The aim of the study is clearly stated - to situate prone burials within broader medieval and post-medieval funerary rite trends, and to close the research gap with a substantial dataset. This has, in general, been achieved.

The authors tackle the use of the term deviant very effectively with Sonderbestattung, which is wholly appropriate and mirrors the use of special or structured deposits described by zooarchaeologists working with atypical placements of animal remains. Subsequently deviant is used throughout the paper with this caveat.

It is perhaps worth mentioning that most research on deviant burials in Britain has focused on the early medieval period (and particularly in England), whilst late medieval 'deviancy' has been studied comparatively less and has suffered from the same problem as Central European site-based examples, although Gilchrist and Sloane's comprehensive survey of monastic cemeteries is the best study to date of the diversity of late medieval funerary rites. This has not been cited and it would be worth including (Requiem: The Medieval Monastic Cemetery).

Throughout the paper the use of chronological segments is effective in relation to highlighting the major trends observed, however I would urge the authors to clarify some of their terms in the earlier sections. Chronological designations such as "advanced early middle ages" are not very helpful, and it would be better to consistently use the same chronological time slices or specify centuries in each case.

I think the parameters that were recorded and statistically compared are useful, and I found the discussion to be articulated in a very accessible way, providing a convincing contextualisation of the statistical trends outlined in the results. There is a good balance between the general categories used for comparison and the level of detail that is included within the discussion, especially for specific temporal and spatial trends. I think the interpretation of a shift from expressions of humility to ritual action taken against harmful revenants is plausible, and the connection with the late dissemination of Slavic "vampire" beliefs (in turn from a likely Balkan origin), as well as the agency of pandemics, is presented in a critical and compelling way.

The figures, in general, are fine, and the tables look ok in their original Excel format. I would suggest for Fig 1, adding an inset map showing the case study regions within Europe as a whole.

6. PLOS authors have the option to publish the peer review history of their article (what does this mean?). If published, this will include your full peer review and any attached files.

Reviewer #1: No

Reviewer #2: No

---

## [Author Response · Author response to Decision Letter 0]

5 Aug 2020

First of all, we would like to thank the two reviewers and the editor for their valuable comments and constructive criticism. The manuscript underwent a language check by a native speaker.

We have integrated the followings comments in the manuscript.

1. Style requirements, tables and supporting information

Style of the title page and the main body were adapted to PLOS ONE’s style requirements. Tables were inserted in the manuscript and information on the Supporting information files was added.

2. Additional information on specimens

The study deals with archaeological specimens, of which the majority is published. We clarified that the specimens are stored in the respective heritage institutions of the states or cantons. In Table 1, we added the states/cantons to the countries. The statement „All necessary permits were obtained for the study, which complied with all relevant regulations“ was added to the material section.

3. Map (Figure 1) 

Permission for publication was obtained from the copyright holder of Figure 1 (map). The copyright holder has added information on the map creation: Basic vector map of Europe, the Isohypses are produced by using Copernicus data and information funded by the European Union - EU-DEM layers. The bodies of water are based on data from www.naturalearthdata.com . Publication of the basic vector map at Zenodo: http://doi.org/10.5281/zenodo.3457998

4. Age distribution (reviewer #1)

A sentence on the possible bias of the age distribution was added to the discussion. 

5. Geographical bias (reviewer #1)

Sentences have been added on the geographically imbalanced state of research.

6. Volkskunde (reviewer #1)

Indeed, Wiedergänger and Nachzehre are terms of German folklore. We have added expressions to make clear that we summarize the perceptions reported in the Handwörterbuch des deutschen Aberglaubens. In the reference list, we now indicate the original publication date of the book. 

7. Medieval England (reviewer#2)

Two sentences have been added to the research history on deviant burials on the situation in Britain, and Gilchrist and Sloane 2005 have been cited. 

8. Chronology (reviewer#2) 

As requested, we have added centuries to chronological designations, where necessary.

9. Map (reviewer#2)

As requested, we have added a map of Europe to figure 1, indicating our study area.

We are looking forward to your decision and the timely publication of our manuscript.

Yours sincerely,

Amelie Alterauge & Sandra Lösch

---

## [Editor Report · Decision Letter 1]

18 Aug 2020

Between belief and fear - Reinterpreting prone burials during the Middle Ages and early modern period in German-speaking Europe

PONE-D-20-13088R1

Dear Dr. Lösch,

We’re pleased to inform you that your manuscript has been judged scientifically suitable for publication and will be formally accepted for publication once it meets all outstanding technical requirements.

Kind regards,

Peter F. Biehl, PhD

Academic Editor

PLOS ONE
---

## [Editor Report · Acceptance letter]

20 Aug 2020

PONE-D-20-13088R1 

Between belief and fear - Reinterpreting prone burials during the Middle Ages and early modern period in German-speaking Europe 

Dear Dr. Lösch:

I'm pleased to inform you that your manuscript has been deemed suitable for publication in PLOS ONE. Congratulations! Your manuscript is now with our production department. 

Kind regards, 

on behalf of

Dr. Peter F. Biehl 

Academic Editor

PLOS ONE